# OCT4 impedes cell fate redirection by the melanocyte lineage master regulator MITF in mouse ESCs

Danna Sheinboim[1], Itay Maza[2,7], Iris Dror[3,8], Shivang Parikh[1], Vladislav Krupalnik[2], Rachel E. Bell[1], Asaf Zviran[2], Yusuke Suita[4], Ofir Hakim[5], Yael Mandel-Gutfreund[3], Mehdi Khaled[6], Jacob H. Hanna [2] & Carmit Levy[1]

Ectopic expression of lineage master regulators induces transdifferentiation. Whether cell fate transitions can be induced during various developmental stages has not been systemically examined. Here we discover that amongst different developmental stages, mouse embryonic stem cells (mESCs) are resistant to cell fate conversion induced by the melanocyte lineage master regulator MITF. By generating a transgenic system we exhibit that in mESCs, the pluripotency master regulator Oct4, counteracts pro-differentiation induced by Mitf by physical interference with MITF transcriptional activity. We further demonstrate that mESCs must be released from Oct4-maintained pluripotency prior to ectopically induced differentiation. Moreover, Oct4 induction in various differentiated cells represses their lineage identity in vivo. Alongside, chromatin architecture combined with ChIP-seq analysis suggest that Oct4 competes with various lineage master regulators for binding promoters and enhancers. Our analysis reveals pluripotency and transdifferentiation regulatory principles and could open new opportunities in the field of regenerative medicine.

[1] Department of Human Genetics and Biochemistry, Sackler Faculty of Medicine, Tel Aviv University, Tel Aviv 69978, Israel. [2] Department of Molecular Genetics, Weizmann Institute of Science, Rehovot 7610001, Israel. [3] Department of Biology, Technion–Israel Institute of Technology, Haifa 32000, Israel. [4] Cutaneous Biology Research Center, Department of Dermatology and MGH Cancer Center, Massachusetts General Hospital, Boston, MA 02114, USA. [5] The Mina and Everard Goodman Faculty of Life Sciences Bar-Ilan University, Ramat Gan 5290002, Israel. [6] Institut Gustave Roussy, INSERM 1186, Villejuif 94805, France. [7] Present address: Department of Gastroenterology, Rambam Health Care Campus & Bruce Rappaport School of Medicine, Technion Institute of Technology, Haifa 31096, Israel. [8] Present address: Department of Biological Chemistry, University of California, Los Angeles, CA 90095, USA. Danna Sheinboim, Itay Maza, Jacob H. Hanna and Carmit Levy contributed equally to this work. Correspondence and requests for materials should be addressed to J.H.H. (email: jacob.hanna@weizmann.ac.il) or to C.L. (email: carmitlevy@post.tau.ac.il)

An ultimate goal of regenerative medicine is to produce functional differentiated cells suitable for transplantation. For this purpose, two main reprogramming approaches are currently available. One reprograms somatic cells into induced pluripotent stem cells (iPSCs) by the induction of the four Yamanaka factors[1–3] and subsequently differentiates them into the desired somatic cells. The other is transdifferentiation, which is the direct conversion of one somatic cell type to another without going through pluripotency, by the manipulation of one or more ectopic master regulator transcription factors[4, 5]. Both of these reprogramming approaches face critical challenges of producing optimized cultures of reprogrammed target cells with high level of efficiency and quality. In order to meet these challenges a fully controlled directed differentiation and transdifferentiation is needed.

To date numerous studies have manipulated stem cell differentiation and cell fate redirection by cell culture growth media and microenvironment conditions[6]. However, it remains to be known whether pluripotent stem cells (ESCs or iPSCs) would be directly differentiated by inducing expression of a lineage specific master regulator that is known to successfully redirect cell fate in somatic cells.

ESCs are pluripotent cells derived from the inner cell mass of blastocyst-stage embryos that have the capacity to give rise to differentiated derivatives from all three primary germ layers: ectoderm, mesoderm, and endoderm[7]. iPSCs share chromatin structure and gene expression characteristics with ESCs[8]. ESCs differentiation into specialized cells requires the use of differentiation media that contains specific growth and signaling factors per se or in combination with the expression of defined

transcription factors[6]. This approach has been used to differentiate ESCs into melanocytes[9–11].

Microphthalmia-associated transcription factor (MITF) is a basic helix–loop–helix (bHLH-zip) transcription factor that serves as the master regulator of the melanocyte lineage. Non-functional MITF results in lack of melanocytes[12, 13]. Alternative promoters give rise to various MITF isoforms differing in their N-termini; promoter use is regulated in a tissue-specific manner[14]. The promoters of genes regulated by the melanocyte specific isoform, M-MITF, contain the consensus E-box sequence[12]. MITF regulates the transcription of melanocyte-specific genes: *TRPM1, TYR, TYRP1, DCT, SILV,* and *MLANA* as well as genes involved in cell survival and proliferation such as *BCL2, CDK2,* and *DICER*[15]. MITF has a critical role in melanoma as it is required for survival and controls the proliferative, invasive, and metastatic properties of melanoma cells[16, 17].

It was previously reported that ectopic expression of Mitf converts fibroblasts into cells with melanocyte characteristics, although only a minority of the Mitf-transfected cells had a melanocyte-like cell appearance[18]. A recent study showed that about 10% of the fibroblasts infected with retroviruses carrying a combination of transcription factors MITF, SOX10, and PAX3 acquire melanocyte properties[19].

In an effort to produce highly efficient, directed differentiation and transdifferentiation systems, we analyze the effects of inducible expression of the Mitf master regulator in differentiation promoting growth conditions in both somatic cells and pluripotent ESCs. We develop a transgenic system in which Mitf expression is Doxycycline (Dox) inducible that enable us to screen Mitf reprogramming potential on different primary cell

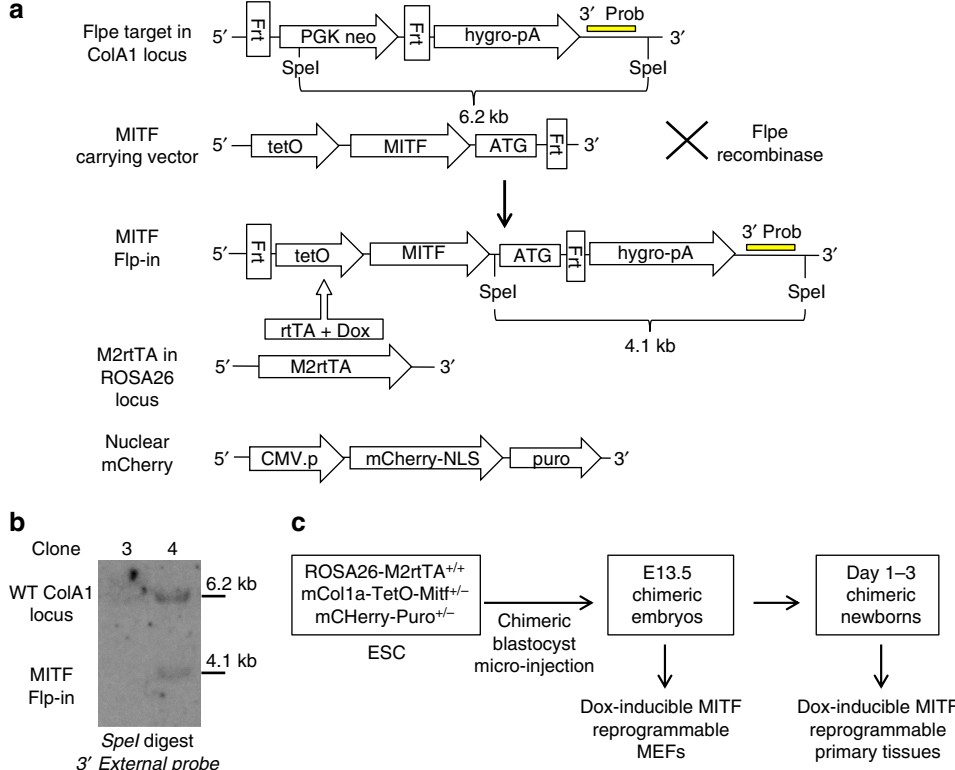

**Fig. 1** Generation of Dox inducible Mitf reprogrammable system. **a** A schematic illustration of the Frt-tetO-Mitf knock-in construct directed to the Flpe mCol1a cassette. Also indicated are 3′ Southern blot external probes and the nuclear mCherry transgene marker. **b** Southern blot verification of tetO-Mitf knock-in to mESCs. Clone #3 was incorrectly targeted, whereas clone #4 was correctly targeted and was used for injection into E3.5blastocyst to obtain ROSA26-M2RtTa⁺/⁺mCol1a-TetO-*Mitf*⁺/⁻ derived MEFs and somatic cells. **c** A schematic illustration of the Mitf knock-in reprogrammable MEFs and somatic cells

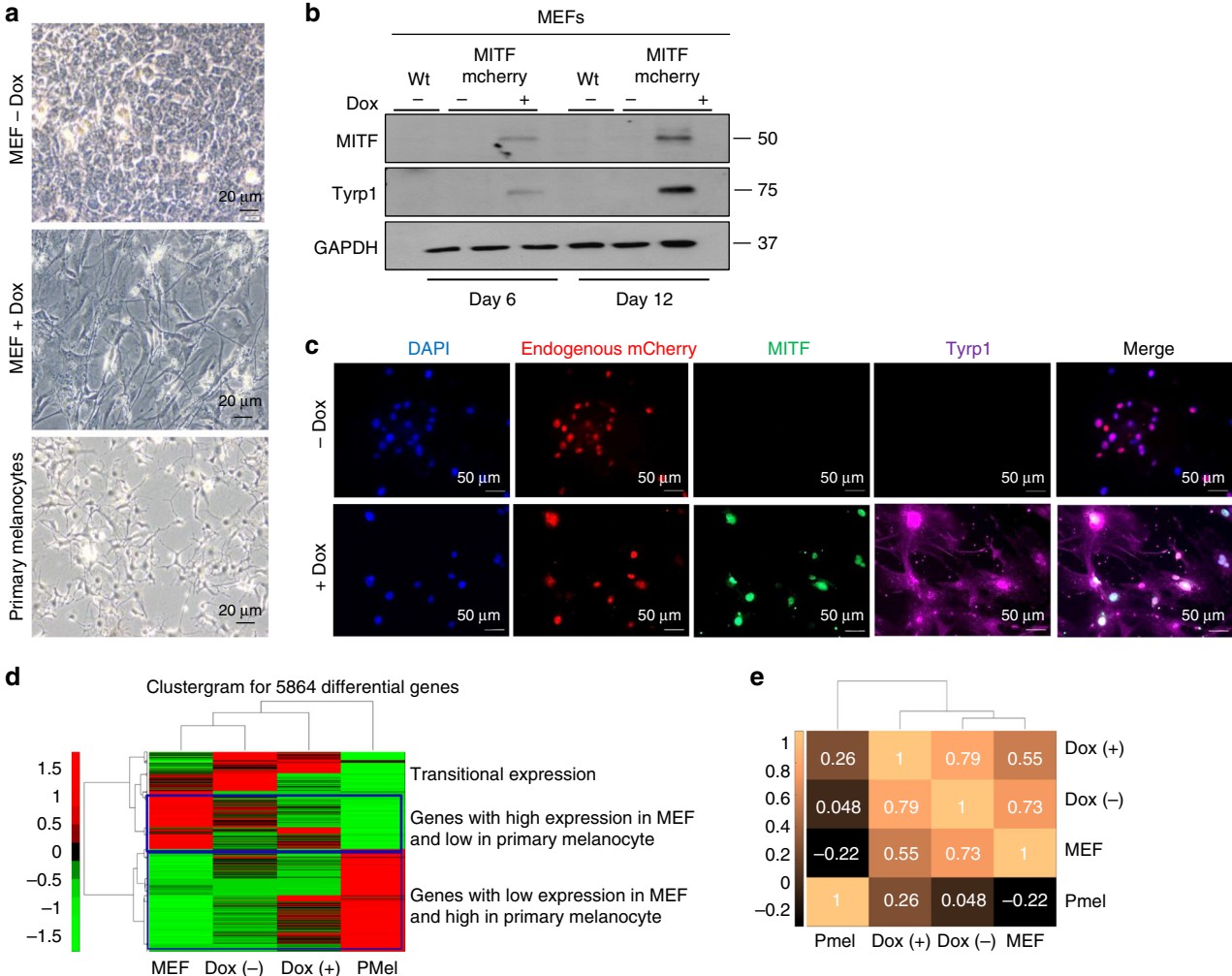

**Fig. 2** MITF efficiently transdifferentiates MEFs. **a** Bright field images of Mitf knock-in MEFs showing morphological changes at 12 days after Mitf induction by Dox supplementation and of untreated Mitf knock-in MEFs and primary mouse melanocytes. This was one of $n = 3$ experiments. **b** MITF and TYRP1 protein levels in Mitf knock-in MEFs at days 6 and 12 post Dox induction. This was one of $n = 3$ experiments. **c** Immunostaining of MITF (green) and TYRP1 (purple) in MEFs at day 6 post Dox induction with endogenous expression of mCherry (red). This was one of $n = 3$ experiments. **d** Hierarchical clustering of genes differentially expressed in MEFs before and after induction of Mitf expression (5864 genes) using Spearman correlation as a distance metric, Ward's linkage, and per-row standardization (z score). **e** Spearman's correlation was calculated over all exons that show differential ( > 4 fold change) expression between MEFs and melanocyte samples (45,155 exons)

types originating from the same animal model and also to direct differentiation of pluripotent mESCs.

## Results

**Generation of Dox inducible Mitf reprogrammable system.** In order to test the potential of different cell types to switch lineage in response to a single lineage master regulator expression, we established a Dox-inducible mouse model using the melanocyte lineage specific isoform of Mitf (M-Mitf). M-Mitf serves here as an exclusive inducer of differentiation and transdifferentiation unlike the conventional method based on media promoting factors. We used KH2 mESCs that constitutively express the M2 reverse tetracycline transactivator (M2rtTA)[20] from the *ROSA26* locus (*ROSA26*-M2rtTA). A Dox-responsive element controlling Mitf expression was targeted downstream of the *collagen 1a1* locus by frt/Flpase-mediated site-specific integration (*Col1a1-TetO-Mitf*). In order to visualize the engineered mESCs, pBRY-CAGGS-mCherry encoding transgene was transduced into the correctly targeted mESCs (Fig. 1a, b). mESCs were injected into host E3.5 mouse blastocysts and chimeric embryos were dissected

at 13.5E and were used to generate mouse embryonic fibroblast cultures (MEFs). Chimeric newborn P1-3 mice were also generated and somatic tissues were analyzed (Fig. 1c). The constitutively expressed mCherry allele and puromycin resistance cassettes allowed us to easily isolate and purify the Mitf transgenic cells from chimeric cultures and animals.

**MITF efficiently transdifferentiates MEFs.** Somatic cell transdifferentiation into melanocyte-like cells has been previously reported in MEFs at efficiencies of 2 to 10%[18, 19]. Although transdifferentiation is considered to be superior in terms of reprogramming efficiency as compared to conventional iPSCs reprogramming, the acceptable range of transdifferentiation protocols is up to 20% efficiency of the source cell population[21–23].

We first aimed to validate the ability of Dox-induced Mitf expression to stimulate transdifferentiation in mouse cells. We generated MEFs from the engineered mouse chimera embryos and induced Mitf expression by supplementing growth media with Dox. Dox treated Mitf knock-in MEFs acquired a dendrite like morphology that is a distinctive characteristic of

melanocytes[24] (Fig. 2a) whereas Mitf knock-in mESCs appearance remained unchanged (Supplementary Fig. 1a).

We then examined expression of known MITF target genes upon Dox induction. Mitf knock-in MEFs showed higher expression of melanogenic markers at 6 days post Dox induction than at earlier time points (Supplementary Fig. 1b, c). We therefore treated Mitf knock-in MEF and mESCs cells for 6 days

and demonstrated upregulation of MITF target genes, TYRP1 and TYR (Fig. 2b and Supplementary Fig. 1d). Moreover, TYRP1 upregulation in Dox-treated MEFs compared to the vehicle-treated MEFs was demonstrated by immunostaining (Fig. 2c) and quantified (Supplementary Fig. 1e, f). MITF, as an expected of a transcription factor, was localized to the nucleus, whereas TYRP1 was found in the cytoplasm since it is part of the secreted

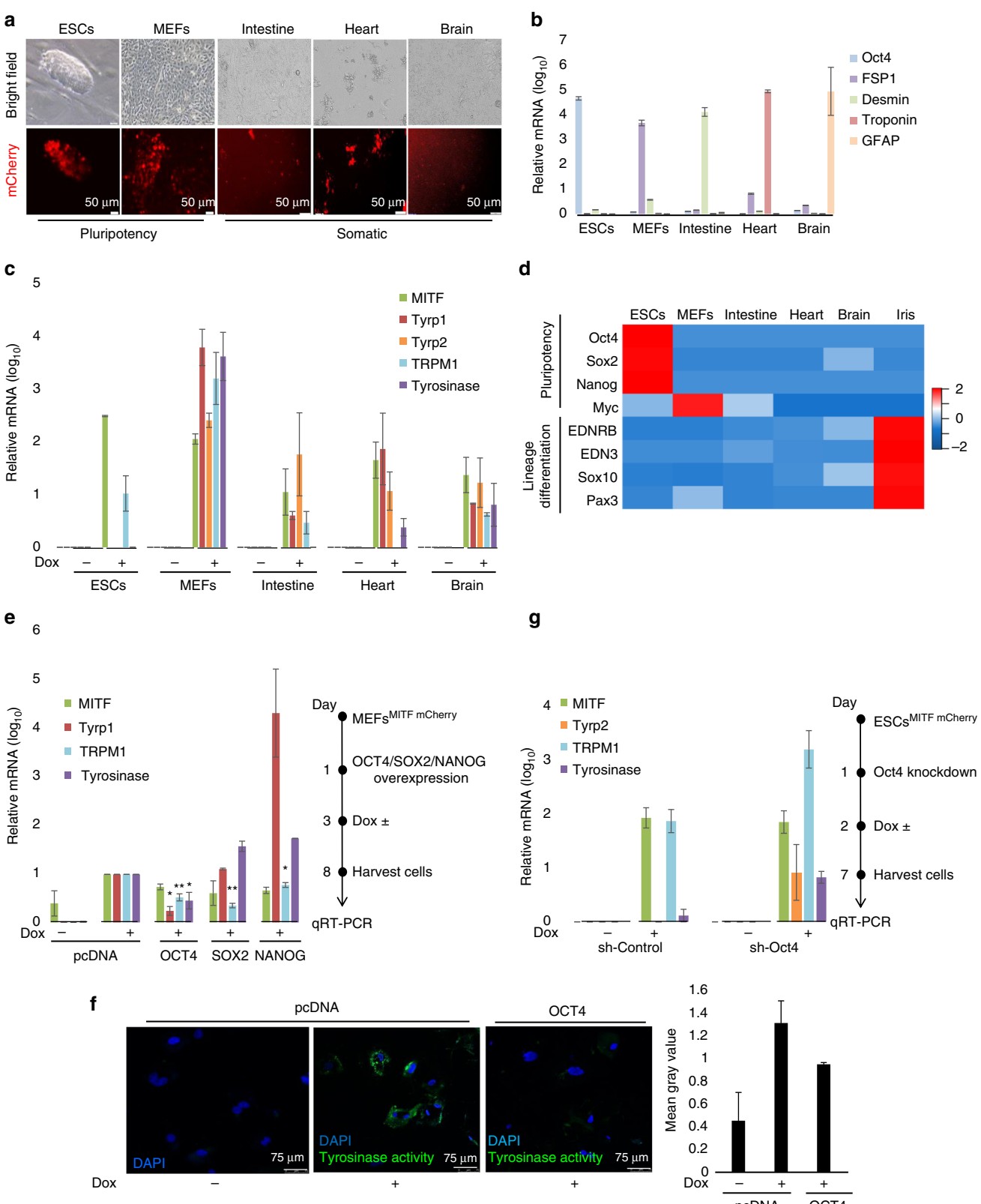

pigment-producing vesicles[25] that are trafficked from melano-cytes to adjacent cells[26]. Finally, to examine the global effect on the transcriptome, we performed a global mRNA expression profiling via RNA-seq. Genes differentially expressed in Mitf knock-in MEFs compared to vehicle-treated knock-in MEFs (Supplementary Data 1; Supplementary Table 1) were clustered and compared to expression in untreated wild-type MEFs and primary mouse melanocytes (PMel) (Fig. 2d). This clustering demonstrates a gradual transition in the expression profile of Dox treated Mitf knock-in MEFs toward the melanocyte expression profile (Fig. 2d). Likewise, Spearman correlation coefficients clearly demonstrate that Dox treated Mitf knock-in MEFs (Dox + ) are more similar to primary melanocytes (0.26) than to untreated Mitf knock-in MEFs (0.048), but overall Dox-treated knock-in MEFs are still more similar to wild-type MEFs (0.73) than to primary melanocytes (Fig. 2e). Indicating that Dox + are MEFs that started to change upon primary melanocytes, but on the imaginary differentiation trajectory they are still closer to untreated MEFs. In summary, the Mitf inducible transgenic system recapitulated the previously reported transdifferentiation ability of MITF in mouse fibroblasts[18]. The higher efficiency is likely attributed to an optimized transgenic delivery of the reprogramming factor and to higher levels of factor induction than obtained during heterogeneous primary transduction with viruses[27] (Fig. 2c).

**OCT4 impedes mESCs differentiation despite MITF expression**. Successful transdifferentiation by MITF in somatic cells encouraged us to further investigate our hypothesis in ESCs. Since naïve ESCs are pluripotent and their euchromatin to hetero-chromatin ratio is higher than in somatic cells[28], we reasoned that ESCs might be favorable for manipulations, thereby enabling directed differentiation by MITF induction. In order to test this assumption, in addition to Mitf knock-in mESCs and Mitf knock-in MEFs, we also generated primary cell cultures from the intestine, heart, and brain of Mitf knock-in mouse chimeric newborns as these tissues represent the three germ layers endo-derm, mesoderm, and ectoderm, respectively (Fig. 3a). We vali-dated the quality of cell isolation by measuring expression of lineage specific markers (Fig. 3b). This analysis indicated high purity of isolation for each tissue. Next, Mitf expression was induced in all of the indicated preparations by addition of Dox to the growth media for 6 days. Interestingly, Mitf was upregulated in all cell types, however, Mitf target genes were not upregulated in mESCs, although these genes were upregulated in other tested somatic cell types (Fig. 3c).

In order to elucidate why mESCs were less responsive to Mitf induction, we profiled gene expression using the "BioGPS" gene annotation portal (Supplementary Data 2). Our first speculation was that mESCs, unlike the other cells, lack the expression of melanocyte lineage developmental genes such as Ednrb, Edn3, Sox10, and Pax3[13]; however, all of the investigated cells exhibited comparable expression amounts of melanocyte lineage develop-mental genes (Fig. 3d). This suggested that there are factors that prevent mESCs differentiation upon Mitf induction.

Possible candidates are OCT4, SOX2, and NANOG, which are the core factors that maintain the pluripotent state and which play vital roles in the control of specific cell fates dependent on their defined levels[29–31]. In order to investigate whether one or more of these factors block Mitf from upregulating its target genes, we first overexpressed OCT4, SOX2, or NANOG in MEFs followed by Mitf induction. Interestingly, although Mitf was significantly induced, there was only minor upregulation of its specific target genes Tyrp1, Trpm1, and Tyrosinase when OCT4 was overexpressed (Fig. 3e). SOX2 overexpression blocked Trpm1 upregulation but not induction of Tyrp1 and Tyrosinase, suggesting a more dominant role for OCT4. OCT4 overexpres-sion (confirmed as shown in Supplementary Fig. 2) in Dox treated Mitf knock in MEFs also inhibited TYR activity (Fig. 3f) as shown using a florescent dye bound to the tyrosine analog tyramide, which is a substrate for TYR[32]. In a reciprocal experiment, we inhibited expression of Oct4, using validated shRNA compared to sh-control, followed by Mitf induction in Mitf-inducible mESCs. Samples showed that Mitf upregulates its target genes Tyrp2, Trpm1, and Tyrosinase in mESCs deficient in Oct4 (Fig. 3g).

In summary, the above data indicate that OCT4 impedes Mitf activity in mESCs and in somatic cells upon ectopic expression of OCT4. This suggests that OCT4 blocks MITF pro-differentiation activity. We therefore reasoned that for a highly efficient directed differentiation of mESCs by Mitf induction, similarly to Mitf induced transdifferentiation of MEFs, a concomitant Oct4 suppression is required in order to release the stem cells from pluripotency.

**OCT4 interferes with MITF transcriptional activity**. To explore the mechanism underlying OCT4 interference with the pro-differentiation activity of MITF, we examined OCT4 and MITF genomic occupancy by analysis of ChIP-seq data of MITF in human melanocytes[33] and OCT4 in human ESCs[34]. We found that 26% of promoters bound by MITF were also bound by OCT4, and in 13% of these regions OCT4 bound within 10 Kb of the MITF-bound region (Fig. 4a). Of regions bound by MITF, 39% were also bound by OCT4 in a close proximity of less than 3Kb (Fig. 4b). For instance, regions upstream of TRPM1 and TYR, genes strongly related to melanocyte identity, were occupied by both MITF and OCT4 (Fig. 4c).

Next, we tested the capability of OCT4 to influence MITF transcriptional activity by using reporter constructs in which luciferase expression was driven by the TRPM1 or TYR promoter. TRPM1 and TYR promoter regions contained not only the well-documented MITF binding consensus site (the E-box)[12] but also

**Fig. 3** OCT4 impedes mESCs differentiation despite MITF expression. **a** Representative microscopy images of mESCs, MEFs, and cells from the intestine, heart, and brain from Mitf knock-in chimera mice. This was one of $n = 3$ experiments. **b** Expression of lineage specific markers in the investigated cell types are shown. Levels were normalized to Gapdh. Error bars represent $\pm$ SEM ($n = 3$). **c** Mitf, Tyrp1, Tyrp2, Trpm1, and Tyrosinase mRNA levels in the indicated cells at day 6 post Dox induction and in vehicle-treated cells. Relative levels were normalized to Gapdh. Error bars represent $\pm$ SEM ($n = 3$). **d** Gene expression profile of the investigated cells. **e** MEFs were transfected with expression plasmids or transduced with retroviruses for expression of OCT4, SOX2, or NANOG. Mitf, Tyrp1, Trpm1, and Tyrosinase mRNA levels were evaluated at day 6 post Dox induction. Levels were normalized to Gapdh. Fold changes relative to control cells transfected with empty vector (pcDNA) and treated with Dox are shown. Error bars represent $\pm$ SEM. * indicates $p < 0.05$, ** indicates $p < 0.01$ ($n = 3$). Experiment process is shown schematically to the right. **f** MEFs were transfected with a plasmid for expression of OCT4 or empty vector control (pcDNA). Tyrosinase activity (Cy5, green) was evaluated at day 6 post Dox induction. Nuclei appear blue (DAPI). Green pixel quantification for 10 nuclei from each treatment using ImageJ software is plotted to the right. This was one of $n = 2$ experiments. **g** mESCs were transduced with lentiviral vectors for expression of shRNA targeting Oct4 or empty vector as control (sh-control). Mitf, Tyrp2, Trpm1, and Tyrosinase mRNA levels were evaluated at day 6 post Dox induction. Levels were normalized to Gapdh, and fold changes relative to control are shown. Error bars represent $\pm$ SEM ($n = 2$). Experiment process is presented schematically on the right

the known OCT4 binding motif[35] (Fig. 4d, upper panel). We compared effects of wild-type OCT4 to that of a mutated OCT4 which was phosphorylated in a site located within the OCT4 homeobox domain (T234 and S235) and found to negatively regulate OCT4 by disrupting sequence-specific DNA binding[36]. TRPM1 or TYR reporter was co-transfected with an MITF expression vector and wild-type or mutant OCT4 expression plasmid. Cells with high MITF levels, were co-transfected with TRPM1 or TYR reporters and wild-type or mutant OCT4

expression vector. In both systems, wild-type OCT4 significantly decreased MITF transcriptional activation of TRPM1 and TYR whereas mutant OCT4 did not (Fig. 4d). OCT4 also inhibited MITF-induced transcription of mouse MLANA reporter (Supplementary Fig. 3a). SOX2 and NANOG overexpression in these systems did not inhibit expression from the reporters as OCT4 did (Supplementary Fig. 3b). When melanoma cells with high MITF levels were transfected with a vector for expression of OCT4 there was a reduction of both TRPM1 and TYR expression

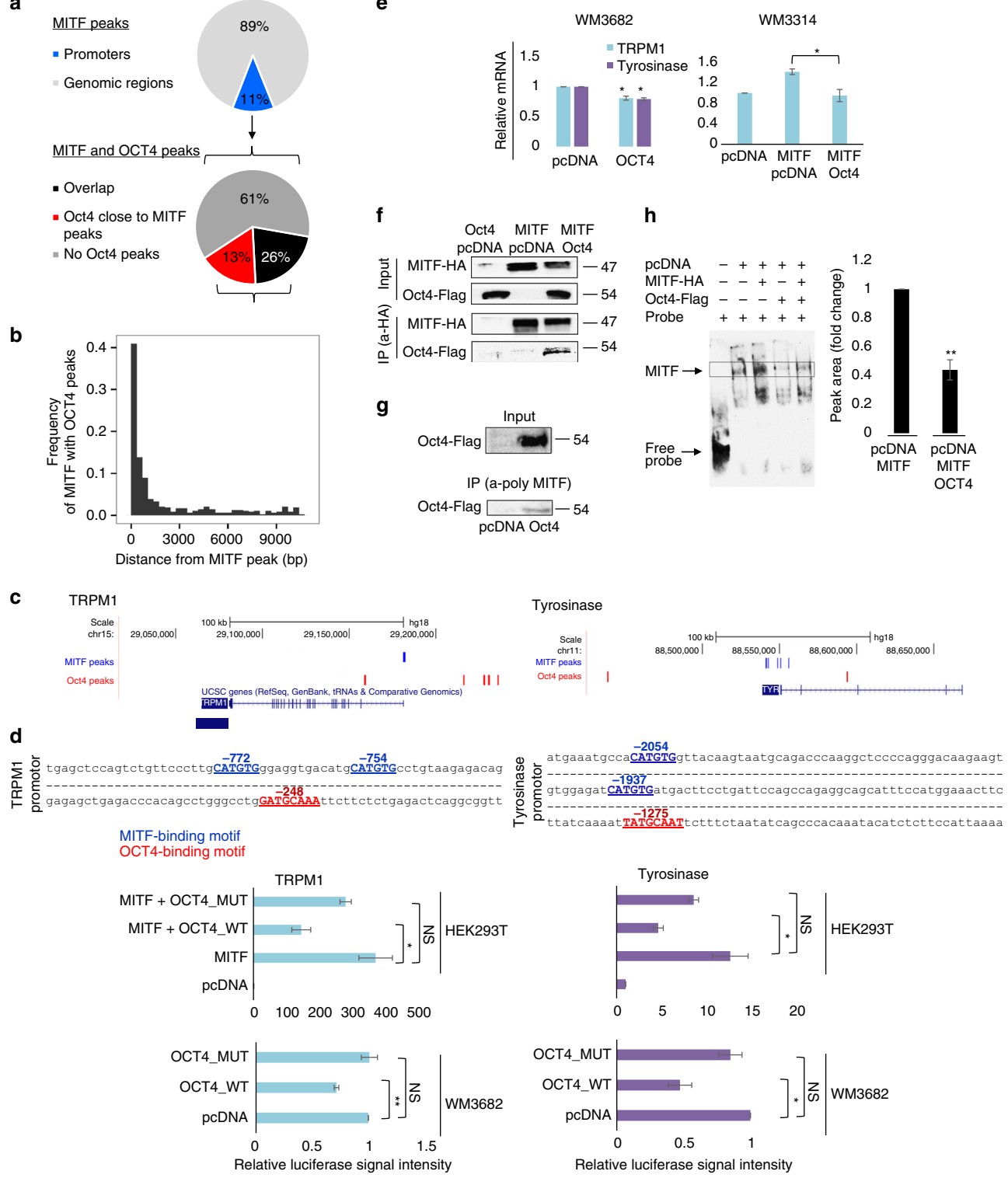

(Fig. 4e). Furthermore, melanoma cells with low MITF levels that were transfected with vectors for expression of MITF and OCT4 showed a significant decrease in TRPM1 mRNA levels compared to the control (Fig. 4e). Interestingly, OCT4 and MITF are inversely correlated in different melanoma cell lines (Supplementary Fig. 3d) and seem to be uniquely expressed in skin cutaneous melanoma (Supplementary Fig. 3e).

To further study the OCT4 mechanism of action in regulating MITF transcriptional activity, Co—Immunoprecipitation (Co-IP) experiments were performed to test their physical interaction. Cells were transfected with vectors for expression of MITF-HA and OCT4-Flag, and HA antibody was used for immunoprecipitation (Fig. 4f). In addition, melanoma cells that endogenously express MITF were transfected with OCT4-Flag, and immunoprecipitation was done using antibody against MITF (Fig. 4g). These experiments demonstrated that OCT4 and MITF physically interact. To confirm that there is a functional interaction between OCT4 and MITF, electrophoretic mobility shift assay (EMSA) experiments were conducted that demonstrated that MITF binding to the E-box consensus site in the TRPM1 promoter was blocked by OCT4 (Fig. 4h; Supplementary Fig. 3c). Interestingly, Co-IP results indicate that MITF and OCT4 physically interact, however, a distance was observed between their peaks according to the ChIP-seq analysis, suggesting long-range interactions. Taken together, our data indicate that OCT4 interrupts MITF-mediated transcriptional activity by preventing MITF binding to the promoter region of its putative target genes.

**Oct4 maintains pluripotency and prevents differentiation**. Next, we aimed to explore whether OCT4 interference with MITF transcriptional activity is a global phenomenon and whether other somatic lineage commitment transcription factors are subjected to such regulation. Therefore, we analyzed ChIP-seq data of OCT4 and the following lineage transcription factors: MITF[33], CDX2, GATA4[37] and HES1[38], which are central in the differentiation of melanocytes, intestinal cells, cardiac cells, and astrocytes, respectively (Fig. 5a). There was significant overlap of genes bound by OCT4 and these transcription factors (-log10 P-values for the overlap between OCT4 and MITF, 222.53; HES1, 177.38; GATA4, 235.73; CDX2, 193.11). Using the DAVID functional annotation tool, we performed a Gene Ontology (GO) enrichment analysis of the co-bound genes (Supplementary Data 3). Significance of the obtained biological pathways was determined according to Benjamini-Hochberg multiple testing rate. The pathways that were considered significant scored below 0.05 in this parameter. The identified biological pathways clearly showed that OCT4 occupies promoters of genes that are essential for specific lineage development (Fig. 5b). For example, GO

analysis of genes co-bound by OCT4 and MITF revealed significant enrichment of the melanin pigmentation process, which uniquely characterizes melanocytes and is also known to be regulated by MITF[39]. Similarly, analysis of OCT4 and GATA4 bound genes revealed significant enrichment in genes involved in cardiac development, which is known to be regulated by GATA4[40, 41]. Importantly, the non-overlapping genes yielded only general biological pathways and not genes known to be regulated by the lineage-specific transcription factors (Supplementary Data 4).

In order to explore whether OCT4 uniquely occupies promoters of genes involved in developmental lineage pathways, we analyzed ChIP-seq data of the general transcription factor E2F7 for overlap with MITF-bound genes. In the GO analysis of co-bound genes we found no significant enrichment in developmental pathways (Supplementary Data 5; Supplementary Fig. 4a, b). Additional analyses of MITF ChIP-seq data generated in normal melanocytes[42] with ChIP-seq peaks of OCT4, E2F7, or P53, followed by GO analysis of overlapping genes (Supplementary Data 5) showed that only MITF and OCT4 co-bound genes resulted in significant melanocyte differentiation pathway enrichment (Supplementary Fig. 4c, d). No enrichment in genes involved in differentiation was observed in the genes bound by MITF and E2F7 or P53. No other classes of genes were significantly enriched in these overlapping targets.

These data suggest that OCT4 occupies lineage-specific gene promoters in order to prevent transcriptional activity under inappropriate timing. To test this hypothesis in vivo, we used mice in which Oct4 expression was inducible[43]. We derived transgenic melanocytes and cell cultures from the brain, heart, and intestine of newborns (Fig. 5c). The expression of cell type specific markers was significantly decreased following Oct4 induction (Fig. 5d).

We also examined OCT4 in relation to other transcription factors in another regulatory layer of chromosome organization. The Hi-C method was developed to enable exploration of genome-wide chromosomal associations[44]. Segments of internal high spatial connectivity termed "topologically associating domains" (TADs)[45, 46]. Clustering of genes that in TADs are co-regulated in nuclear sub-compartments, which comprise the contact domains, may facilitate their co-regulation by favoring frequent engagement with enhancers within TADs while insulating the influence of the enhancers beyond the TAD borders create nuclear microenvironments that are enriched in specific factors that coordinate the expression or repression of specific groups of genes[47, 48]. Since TAD borders are stable across different cell types[49], we integrated Hi-C data of human lymphoblastoid cells together with ChIP-seq data. This analysis showed a significant sharing domain binding between OCT4 and a number of

**Fig. 4** OCT4 interferes with MITF transcriptional activity. **a** Pie chart represents promoter regions occupied by MITF (upper panel) and of the realms engaged by MITF and OCT4 (lower panel). **b** Frequency of OCT4 and MITF peaks within a given distance. **c** Illustration of MITF and OCT4 peak positions on TRPM1 (left panel) and TYR (right panel) generated by uploading MITF and OCT4 ChIP-seq data[33, 34] into "UCSC genome browser". **d** Upper panel: regions upstream of the transcription start site which were cloned into Luciferase reporter plasmids are shown; numbers indicate nucleotide position. MITF binding motif (E-box: CATGTG) appears blue and OCT4 binding motif (Octamer box: NATGCAAN) appears red. Lower panel: HEK293T or WM3682 cells were co-transfected with luciferase reporter driven by TRPM1 or TYR promoter and wild-type OCT4 (OCT4_WT), mutated OCT4 (OCT4_MUT) or empty plasmid as control. HEK293T cells were also co-transfected with plasmid for expression of MITF. Luciferase activity was normalized to Renilla. Fold changes relative to control are shown. Error bars represent ± SEM, * indicates $p < 0.05$, ** indicates $p < 0.01$ ($n = 3$). **e** Left: WM3682 were transfected with vector for OCT4 expression or empty vector. TRPM1 and TYR mRNA levels normalized to GAPDH. Error bars represent ± SEM, * indicates $p < 0.05$ ($n = 3$). Right: WM3314 cells were transfected with vectors for expression of MITF and OCT4. TRPM1 mRNA levels normalized to GAPDH. Error bars represent ± SEM, * indicates $p < 0.05$ ($n = 3$). **f** Co-IP assay of MITF-HA and OCT4-flag in HEK293T cells. Samples were precipitated using anti-HA antibody. Anti-HA or anti-flag antibodies were used for western blot. This was one of $n = 3$ experiments. **g** Co-IP assay of endogenous MITF and OCT4-flag in WM3682 melanoma. Samples were precipitated using anti MITF antibody. Anti-flag were used for western blot. This was one of $n = 2$ experiments. **h** EMSA was conducted using a probe corresponding to the E-box region. HEK293T nuclear extracts were incubated with biotinylated probe. Bands corresponding to MITF binding and free probe are marked with arrows. Graph represent bands quantification ± SEM, * indicates $p < 0.05$ ($n = 3$)

transcription factors in TADs (Fig. 5e). Assuming that the loci bound by transcription factors are connected in three dimensions to genes within the same TAD, the enrichment of particular transcription factors in the same TAD suggests that they co-regulate genes found in the TAD. To visually demonstrate our analysis, we used "Juicebox" software for visualizing data from Hi-C mapping experiments. By this analysis, for example, the MITF target gene *TRPM1* was found to be in an OCT4 and MITF

shared TAD (Fig. 5f), supporting our hypothesis that OCT4 interferes with MITF transcriptional activity by sharing the same TADs (Supplementary Data 6). These findings indicate that cell type-specific regulatory sites can be engaged by a combination of transcription factors in a cooperative manner.

Finally, we analyzed the enhancers occupied by MITF and determined whether OCT4 or P53 co-bind these regions (Supplementary Data 7). We counted the number of MITF,

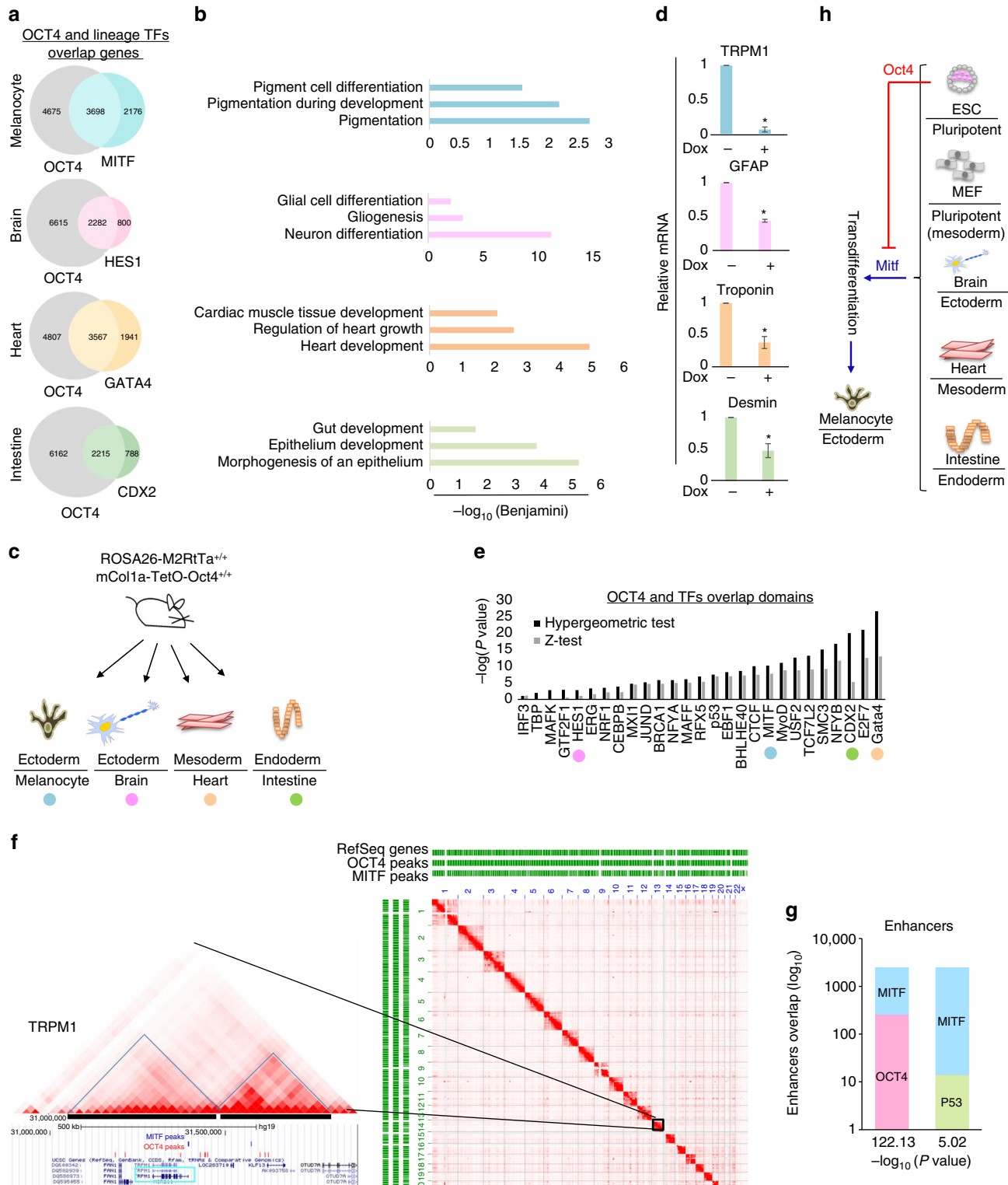

OCT4, and P53 ChIP-seq peaks within ESCs enhancers. We found that 1.6% of enhancers are bound by MITF and 11% of MITF-bound enhancers are also bound by OCT4 (-log($P$-value = 122)). In contrast, only 0.57% of MITF-bound enhancers are also bound by $P53$ (-log($P$-value = 5)) (Fig. 5g). This result implies that the regulatory interaction between MITF and OCT4 occurs in both enhancers and promotors. This strengthens our hypothesis that there is a transcriptional regulatory competition between MITF and OCT4.

## Discussion

Transdifferentiation of somatic cells can be achieved by the expression of a specific set of transcription factors. It has been reported that the expression of MITF alone or in combination with other transcription factors can induce transdifferentiation of fibroblasts into cells with melanocyte characteristics[18, 19]. In this study, we first showed that MEFs can be induced to transition into melanocyte-like cells with high efficiency using an optimized Dox dependent Mitf knock-in system. We also demonstrated that MITF was able to transdifferentiate the investigated somatic cells. Our findings confirm the role of MITF as a master regulator of melanocytes, reminiscent of the master role of MyoD in inducing fibroblast transdifferentiation into muscle[50].

To date, most protocols designed to induce changes in cell fate have focused on reprogramming of a specific cell type[4, 5] and these protocols have not been systemically characterized[50]. Here we evaluated a repertoire of primary somatic and pluripotent cells that originated from the same mouse model and subjected them to the induction of a master differentiation and transdifferentiation regulator, MITF. In contrast to previous reports of transdifferentiation in which indicated that effects of expression of a master regulator were limited to target cells closely related to the originating cells[4], we observed highly efficient induction of transdifferentiation that was not restricted to a specific cell type. We demonstrated transdifferentiation of cells from all three germ layers, however, mESCs did not exhibit a transition into melanocyte-like cells upon Mitf induction.

Since ESCs possess a more open chromatin configuration than somatic cells and thus have higher developmental plasticity, we expected that mESCs would be more susceptible to transcription factor occupancy than somatic cells. Interestingly, although Mitf was induced at high levels in mESCs, it did not promote efficient differentiation into melanocytes. In order to identify the factor responsible for blocking Mitf-induced differentiation of mESCs, we focused on the pluripotent master regulators OCT4, SOX2, and NANOG since all are well established to control pluripotent cell fate decisions[29–31]. This suggests that OCT4 may inhibit the induction of differentiation by MITF in pluripotent cells by physical interaction with MITF, thus interfering MITF from regulating its target genes.

The role of OCT4 in reprogramming of somatic cells into pluripotent stem cells has been widely studied[1]. Recently, Chronis et al. showed that an important step in this process requires the silencing of MEFs enhancers and promoters by OCT4, SOX2, KLF4, and cMYC. Interestingly, these transcription factors induce the relocation of somatic transcription factors from the MEF enhancers, emphasizing their role in blocking somatic transcription factor-dependent differentiation[51]. Computational analysis demonstrated that OCT4 occupies the promoter and enhancer regions of MITF target genes in melanoma cells and melanocytes and also shares chromatin contact domains with MITF. In contrast, no lineage-related pathways were enriched in GO analyses of genes that are bound by MITF and the general transcription factors E2F7 or P53. SOX2 and NANOG also appear to compete for binding to gene promoters with lineage-specific transcription factors in a similar manner as OCT4 as demonstrated by significant GO enrichment for lineage pathways (Supplementary Data 8 and 9). Interestingly, when comparing OCT4[34], SOX2, and NANOG promoter occupancies[52], OCT4 and NANOG together seem to engage more gene promoters than combinations of OCT4 and SOX2 or SOX2 and NANOG (Supplementary Fig. 4e). OCT4 also occupies gene promoter regions and topological domains of somatic lineage transcription factors target genes as well.

Taken together our observations strongly suggest that the ability of Mitf to differentiate pluripotent mESCs is tempered by Oct4 interference (Fig. 5h) at the transcriptional level. The expression of MITF, which serves as a model for other lineage master regulators, efficiently switched cell fate of inter germ layers, thus may open new opportunities in the field of regenerative medicine. It will be of interest to validate a role for OCT4 in human somatic cell reprogramming as well.

## Methods

**Mice**. Dox inducible Oct4 mice: The Oct-4/rtTA (ROSA26-M2rtTA;Col1a1-tetO-Oct4) mice were purchased from the Jackson Laboratory (JAX Stock# 006911). Dox inducible Mitf mice: To enable efficient generation of targeted mESCs harboring a single Dox regulated Mitf gene, we adapted the *ColA1* Flp/FRT recombinase-mediated targeting system previously described by Jaenisch and colleagues[20]. Specifically, we have used mouse melanocyte specific isoform, M-Mitf, flanked by flippase recognition target (FRT) sites. Pre-engineered 'KH2' ESCs contain a FRT-hygro-pA "homing" cassette downstream of the *ColA1* gene (CHC), as well as a reverse tet-transactivator (rtTA) targeted to the *ROSA26* locus (R26-rtTA). Following co-electroporation of pCAGs-Flpe and an appropriate targeting vector, Flpe-mediated recombination between the FRT site at the *ColA1* locus and those present on the targeting vector confers hygromycin resistance only if correctly integrated; by acquiring an initiation ATG codon which is missing in the target locus hygromycin transgene[20]. The resulted mESCs were called *Rosa26* RtTa, *Col1a Mitf*. In order to validate M-Mitf insertion, mESCs were subjected to restriction enzyme (SpeI) treatment followed by DNA extraction and southern blot. M-Mitf in *ColA1* locus represented by a 4.1 KB fragment compared to 6.2 KB fragment in the wild type mESCs (Fig. 2b). In order to create MEFs carrying Dox inducible Mitf, engineered mESCs were microinjected into the inner cell mass (ICM) of BDF2 blastocysts that were implanted into pseudo pregnant female mice. E13.5 or adult chimeric embryos/mice were used to isolate somatic cells. No statistical method was used to predetermine sample size. No animals were excluded from the analysis. The experiments were not randomized. The investigators were not blinded to allocation during experiments and outcome assessment. All mouse animal experiments were approved by Weizmann Institute IACUC (approval # 33550117-2 and 33520117-3).

---

**Fig. 5** Oct4 maintains pluripotency and prevents differentiation. **a** Venn diagrams represent the number of gene promoters bound by OCT4 and lineage specific transcription factors based on ChIP-seq. **b** A view of the selected biological pathways identified by Gene Ontology enrichment analysis of genes bound by OCT4 and lineage-specific factors. **c** Melanocytes were generated from the epidermis and primary cells were extracted from brain, heart, and intestine of mice that can be induced to express Oct4. **d** Levels of lineage-specific mRNAs in the indicated cell types at day 6 post Dox induction of Oct4 expression relative to vehicle treated cells are shown. Levels were normalized to *Gapdh*. Error bars represent ± SEM. * indicates $p < 0.05$ ($n = 2$). **e** Overlap between OCT4 and indicated transcription factor peaks in Hi-C domains. Lineage specific transcription factors are highlighted in colors corresponding to the Venn diagrams in **a**. **f** Right: Hi-C map of human lymphoblastoid cells. RefSeq genes and ChIP-seq peaks of OCT4 and MITF are shown as green horizontal lines. Left: Chromosome 15 is magnified and the melanogenic marker TRPM1 is marked with blue square. TRPM1 is in the black highlighted domains. **g** Graphs show the number of unique and overlapping enhancer regions bound by MITF and OCT4 or by MITF and P53 based on ChIP-seq coordinates. **h** A suggested model representing that high expression of Oct4 in mESCs inhibits Mitf induced differentiation whereas Mitf induced transdifferentiation is not hindered when Oct4 is found in lower concentrations

**Tissue culture**. For generation of MEFs, 13.5E embryos were freshly isolated and their body was enzymatically and mechanically dissociated. WM3314 and WM3682 melanoma cells were kindly given by Dr. Levi A. Garraway (Department of Medical Oncology and Center for Cancer Genome Discovery, Dana-Farber Cancer Institute, Boston, MA). Cells were cultured in DMEM medium supplemented with 10% fetal bovine serum (FBS) (Sigma-Aldrich), 2 mM l-glutamine and 1% penicillin/streptomycin/glutamine (Invitrogen). For generation of mouse mESCs, E3.5 blastocysts were collected and the ICM were dissected mechanically. The cells were plated on irradiated feeder layers (e.g. MEFs) seeded on gelatin-coated 6-well plates and kept in ESCs medium [DMEM with 20% FBS (Invitrogen), 1 × nonessential amino acids, 2 mM l-glutamine, 0.1 mM β-meraptoethanol and 1000 IU/ml leukemia inhibitory factor (LIF)]. After approximately 4–5 days, the outgrowths were trypsinized and dissociated into small cell masses and single cells. These cells were plated on fresh feeders to generate established ES lines.

**Engineered primary somatic cell generation**. Newborn were used at day 1–3. Melanocytes were extracted from the epidermis. Skin was removed into 0.25% trypsin and incubated overnight at 4 °C. Next, dermis and epidermis were separated. Epidermis was incubated at 37 °C in 0.25% trypsin for 30 min and then dispersed into small pieces and covered with melanocyte media. The cells were not touched until day 4 post extraction. Cells from the brain were obtained based on Saura et al[53]. Brains were removed and transferred into Leibovitz's L-15 medium (Biological industries). Then mechanically dissociated with scissors in trypsin B (Biological industries) and incubated at 37 °C for 10 min in heated shaker. 20% medium was added and brains were triturated with glass pipette. Then centrifuged 1400 rpm for 10 min. Pellet was washed with 10% medium and triturated 10 times. Suspension did not touch for 10 min. Then upper supernatant was transferred through 70 um cell strainer (Sigma) and centrifuged 1400 rpm. Pellets were seeded on poly L Lysine (Sigma) covered 6 well plates with fresh DMEM containing 10% FSC, 1% l-glutamine and 1% sodium pyruvate. Medium was changed 1 day post extraction. Cells from heart and intestine were isolated based on Song et al[54]. Heart and intestine were dispersed by addition of 0.625 mg/ml collagenase (type II; Worthington) and incubated at 37 °C for 40 min. Cells were resuspended with fresh DMEM with 10% FCS and centrifuged at 1000 rpm for 3 min. Cell pellet was suspended in DMEM with 10% FCS. Cells originated from the heart were plated on gelatin (Sigma) coated plates.

**RNA Purification and qRT PCR**. Total RNA was purified using Trizol (Invitrogen) according to manufacturer's instructions followed by treatment with RNase-free DNase (QIAGEN). RNA was quantified by measuring OD260/280. For mRNAs analysis, RNA was subjected to one-step qRT-PCR using a MultiScribe qRT-PCR kit (Applied Biosystems) and FastStart Universal SYBR Green Master (Rox) (Roche). mRNA levels were normalized to GAPDH and fold changes relative to control are shown. Error bars represent standard error of the mean ( ± SEM). * indicates $p < 0.05$, ** indicates $p < 0.01$. All qRT-PCR primers are listed in Supplementary Table 2.

**Plasmids and cloning**. hOCT3/4-Flag, hSOX2-Flag and hNANOG-Flag expression vectors were obtained through Addgene. MITF-HA expression vector and pGL3b-hTRPM1-luciferase were kindly obtained from Dr. David. E. Fisher (Department of Dermatology, Cutaneous Biology Research Center, Massachusetts General Hospital, Harvard Medical School, MA). pCEP4_WT_OCT4 and pCE-P4_OCT4_T234E_S235E expression vectors were obtained through Addgene.

**Virus preparation and transduction**. Lentiviral Particles harboring shRNA of mouse Oct3/4, Sox2, Nanog and Control shRNA Lentiviral Particles were purchased from Santa Cruz Biotechnology. MLV particles used for Oct4 and Sox2 overexpression were generated using the pCL-Eco packaging plasmid and pMXs-hOCT3/4 or pMXs-hSOX2 (Addgene through Dr. Jacob Hanna). Both plasmids were co-transfected to HEK293T cells using jetPEI™. 24 h later, viruses were filtered and frozen in −80 °C. Viruses were transduced to mESCs or MEFs using Polybrene ® (Santa Cruz) in final concentration of 1 μg/ml. 24 h later supernatant was replaced to a fresh growth media.

**Southern blot**. Genomic DNA was extracted from each hygromycin-resistant targeted subclone. 10–15 μg of genomic DNA was digested with SpeI restriction enzyme for 5 h and separated by gel electrophoresis. The DNA was transferred to a nitrocellulose membrane that was next hybridized with a radioactively labeled probe and developed using enhanced chemiluminescent substrate (Thermo Scientific).

**Transfection and luciferase reporter assay**. Luciferase reporter driven constructs were co-transfected with the pcDNA3-MITF-HA or empty plasmid into HEK293T cells or WM3682 in 24-well plates (total of 1 μg DNA/well) using jet-PEI™ for HEK293T or TransIT-X2® for WM3682 according to the manufacturer's instructions. Cell lysates were prepared 48 h after transfection, followed by activity measurement of FireFly luciferase, using the Dual Luciferase kit (Promega) according to the manufacturer's recommendations. Promoter activity was normalized to the constitutively expressed Renilla or to protein quantity as measured by Bradford reagent. For MEFs transfection TransIT-X2® reagent was used.

**Gel electrophoresis and immunoblotting**. Melanoma cells were lysed 48hr after transfection and MEFs were lysed at day 6 and 12 post Dox induction in buffer containing 50 mM Tris (pH 7.4), 150 mM NaCl, 1% Triton X-100 and protease inhibitor cocktail (Sigma-Aldrich). Samples (40 μg) were resolved by 10% SDS-PAGE, transferred to nitrocellulose membranes, and exposed to the appropriate antibodies: rat monoclonal HA-probe antibody (1:1000, 11867431001; Roche) or mouse monoclonal ANTI-FLAG®M2 antibody (1:1250, F3165; Sigma-Aldrich) or rabbit anti-TYRP1, (1:1000, a gift from the Vincent Hearing lab, NCI) or mouse monoclonal C5 anti MITF, (1:5, kindly provided by Dr. David. E. Fisher, Department of Dermatology, Cutaneous Biology Research Center, Massachusetts General Hospital, Harvard Medical School, MA). Proteins were visualized with SuperSignal Chemiluminescent Substrates (Pierce), using horseradish peroxidase-conjugated secondary antibodies: goat anti-mouse (1:10000, 7076; Cell Signaling; 115-035-003; Jackson), goat anti-rabbit (1:10000, 111-035-003; Jackson) and goat anti-rat, (1:5000, SC-2006; Santa cruz).

**Immunostaining**. MEFs were cultured on glass cover slips (13 mm, 1.5 H; Marienfeld, 0117530), washed three times with PBS and fixated with 4% paraformaldehyde for 10 min at room temperature. Cells were then permeabilized and blocked in 0.1% Triton, 0.1% Tween and 5% FBS in PBS for 15 min at room temperature. Primary antibodies were incubated for 2 h at room temperature and then washed with 0.1% Tween and 1% FBS in PBS three times. Following primary antibodies were used: mouse ANTI-FLAG®M2 (1:1000, Sigma-Aldrich), rabbit anti-TYRP1 (1:1000, a gift from the Vincent Hearing lab, NCI), mouse monoclonal C5 anti MITF (1:50, kindly provided by Dr. David. E. Fisher, Harvard University). Next, cells were incubated with secondary antibody for 1 h at room temperature, washed and counterstained with DAPI, mounted with Shandon Immu-Mount (Thermo Scientific) and imaged. The following secondary antibodies were used: 488 donkey anti-mouse (1:200, 715-545-150; Jackson) and 647 donkey anti-rabbit (1:200, 711-605-152; Jackson).

**Co-Immunopercipitation**. pcDNA3-MITF-HA and pcDNA3-OCT4-Flag were transfected individually or together into HEK293T cells in 10 cm culture plate (total of 10 μg DNA/well). WM3682 cells were transfected with pcDNA3-OCT4-Flag. Cell lysates were prepared 48 h after transfection using IP buffer containing protease inhibitor cocktail (Sigma-Aldrich). Samples were incubated with rat monoclonal HA-probe antibody (Roche) (for HEK293T) or rabbit anti MITF antibody (a gift from the David E Fisher lab, DF/HCC) (for WM3682) overnight at 4 °C. Antibody-antigen complexes were precipitated by incubation of the sample with 30 μl Protein A/G PLUS-Agarose beads (Santa Cruz) for 2 h at 4 °C. Beads were washed 5 times with IP buffer. Then boiled at 95 °C for 5 min × 2 with loading buffer. The Immunoprecipitated material was subsequently investigated by western blot analysis using primary antibodies: rat monoclonal HA-probe antibody (1:1000, Roche) or mouse monoclonal ANTI-FLAG®M2 antibody (1:1250, Sigma-Aldrich). Proteins were visualized with SuperSignal Chemiluminescent Substrates (Pierce), using horseradish peroxidase-conjugated goat anti mouse secondary antibody (1:10000, Cell Signaling) or goat anti rabbit (1:10000, Jackson). All uncropped western blots can be found in Supplementary Figure 5.

**Electrophoretic mobility shift assay**. Nuclear extracts were prepared using a NE-PER nuclear and cytoplasmic extraction kit (Pierce) according to the manufacturer's instructions. The MITF biotin-labeled DNA probes spanning MITF binding sites were obtained from IDT. Binding reactions of 10 μg of nuclear lysates and 0.02 pmol of labeled double-stranded DNA probe were performed for 20 min on ice using a LightShift chemiluminescent EMSA kit (Pierce) according to the manufacturer's instructions. Competition analyses were performed with an excess (30 pmol) of unlabeled probes. Samples were resolved by 5% PAGE in 0.5 × TBE buffer (45 mM Tris borate, 1 mM EDTA) transferred to nylon membranes. Labeled DNA was visualized with the ECL system (Pierce). The super-shift assay is shown in Supplementary Fig. 3c. Probe sequence for the MITF binding site is listed in Supplementary Table 2. The wild-type probe was derived from the TRPM1 promoter, which was shown to bind MITF[55].

**Tyramide based tyrosinase activity assay**. The assay was performed according to Angeletti et al[32]. Briefly, confluent MEFs cell cultures were scraped and washed three times with phosphate-buffered saline (PBS), pH 7.4. Smears of the final cell suspensions were air-dried and tested. Air-dried smears were permeabilized in 0.1% Triton X-100 in PBS for 5 min at room temperature. Tyramide-Cy5 was reconstituted in 50 μL of dimethyl sulfoxide (DMSO), according to the manufacturer's specifications, diluted in 250 μL of Tyramide Signal Amplification (TSA) diluent (Perkin Elmer Life Science Products, Boston, MA), and applied (50 μL/each) to the slides containing permeabilized cells. The slides were then incubated in a humid chamber at room temperature for 20 min, followed by three washes (X3) with PBS and water (X1). Image analysis of the cells was performed on Leica TCS STED (Stimulated Emission Depletion) confocal microscope. Pixels of 10 nuclei were quantified using the Image J software.

**Microscopy and image analysis**. Images were acquired with A1 Axioscope microscope (Carl Zeiss) equipped with DP73 camera (Olympus) or with Z1 Axioscope microscope (Carl Zeiss). Fluorescent and EMSA images were quantitatively analyzed using ImageJ software.

**Enrichment of OCT4 and lineage TFs in Hi C domains**. Genomic topologically associating domains (TADs) in GM12878, as defined by Rao et al. 2014[56], were used for this analysis. Overlapping domains were removed keeping the shortest domains of each region. ChIP-seq data for each transcription factor[57] was used to assign transcription factor binding in each domain. To test for enrichment of both transcription factors peaks within domains we used two methods: 1) Hypergeometric distribution test: hypergeometric score was defined in order to examine the enrichment of OCT4 peaks and a second transcription factor peaks within Hi-C domains. 2) Permutation test: The peaks of the two transcription factors were shuffled between the domains, keeping the original distribution of the number of peaks in different domain lengths. This was done by grouping the domains by their length and shuffling the number of transcription factor peaks between each group. This was done 10,000 times, each time calculating the number of domains containing both transcription factors peaks.

**RNA sequencing and gene expression profiling analysis**. RNA was extracted from Trizol pellets of wild type MEFs, Dox treated Mitf knock-in MEFs for 6 days, untreated Mitf knock-in MEFs and primary mouse melanocytes, and utilized for RNA-seq by TruSeq RNA Sample Preparation Kit v2 (Illumina) according to manufacturer's instruction. DNA sequencing was conducted on Illumina Hiseq1500. Tophat software version 2.0.10 was used to align reads to mouse mm10 reference genome (UCSC, December 2011). Read counts per exon were calculated over all 628,052 exons in mm10 ensemble GTF (UCSC, December 2011), using bedtools coverage command (version 2.16.2). Exons annotated as protein coding, pseudogene or lincRNA ($n = 459,556$) were selected for further analysis. Exon counts were normalized by the exon length in Kbp and by million number of aligned reads per sample, to give RPKM values. Only exons with at least one RPKM call > 10 were selected, resulting in 163,841 exons corresponding to 14,980 genes. Furthermore, exons were filtered to include only exons that show abs (FC) > 4 between MEF and Melanocyte, resulting in 45,155 differentially expressed exons corresponding to 5864 genes. Spearman correlation between samples were done using Matlab (version R2016b) over all exons (Fig. 2e). Gene expression was defined by the maximal expression level (RPKM) of all exons associated to a certain gene. Hierarchical clustering was calculated over all differential genes (5864 genes) Matlab (version R2016b) clustergram command, using Spearman correlation as a distance metric, ward linkage, and per-row standardization (zscore) (Fig. 2d).

**Promoter and Enhancer analysis**. ChIP-seq peaks of each transcription factor were collected from the ENCODE project consortium[57] and assigned to promoters (using different transcription start site (TSS) distances to define promoters) and enhancers (using human embryonic stem cells enhancers as defined by Ernst et al[58] and appears at ENCODE (http://hgdownload.cse.ucsc.edu/goldenPath/hg19/encodeDCC/wgEncodeBroadHmm/). To test for enrichment of two transcription factors within promoters or enhancers we used Hypergeomtric distribution test. We further explored the distance of OCT4 peaks from promoter/enhancer-MITF peaks up to 10 kb.

**Statistics and reproducibility**. Images shown in Figs. 2a and 3a are representative of three independent experiments. Immunoblots shown in Figs. 2b and 4 f–h and Supplementary Fig. 3c are representative of three independent experiments, in Supplementary Fig. 1d are representative of two independent experiments and in Supplementary Fig. 1b were performed once. Immunofluorescent staining depicted in representative images Fig. 2c and Supplementary Fig. 1e are representative of two independent experiments. Immunofluorescent staining for transfection efficiency in Supplementary Fig. 2 are representative of two independent experiments. The experiments were not randomized. The investigators were not blinded to allocation during experiments and outcome assessment. Standard parametric t-test was applied. Standard errors were calculated for each data set. P values < 0.05 were considered as significant.

**Data availability**. The authors declare that all data supporting the findings of this study are available within the article and its Supplementary Information files or from the corresponding author upon reasonable request. RNA-seq data have been deposited in the Genbank database (GEO) under accession code: GSE102333.

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

## Acknowledgements

C.L. thanks the following for grant support: the European Research Council (ERC) under the European Union's Horizon 2020 research and innovation programme (grant agreement No 726225), I-CORE Gene Regulation in Complex Human Disease, Center No. 41/11 and Fritz Thyssen Stiftung. D.S. thanks her supportive family. S.P. thanks Yehudit Auerbach Mehoudar for generous support. J.H.H. is supported by a generous gift from Ilana and Pascal Mantoux, and research grants from the: European Research Council (ERC-2016-COG-726497 and ERC-2015-PoC-692945), Flight Attendant Medical Research Council (FAMRI), Israel Science Foundation (ISF-ICORE, ISF-NFSC, ISF-INCPM & ISF-Morasha programs), Kamin-Yeda program, Minerva fund, Israel Cancer Research Fund (ICRF), Human Frontiers Science Program (HFSP), the Benoziyo Endowment fund, New York Stem Cell Foundation (NYSCF), Kimmel Innovator Research Award, the Helen and Martin Kimmel Institute for Stem Cell Research. J.H.H. is a New York Stem Cell Foundation (NYSCF)–Robertson Investigator.

## Author contributions

D.S. designed the scientific approach, performed biological and molecular experiments and analysis and wrote the manuscript. I.M. designed the scientific approach, performed biological and molecular experiments and analysis and wrote the manuscript. I.D. and Y.M.G. performed all the ChIP-seq analysis, I.D. and O.H. performed the chromosomal domain analysis, Y.S. contributed to the normal melanocyte experiment, S.P. participate at the molecular experiments and analysis, R.E.B. performed the melanoma microarray analysis, M.K. contributed to the scientific hypothesis and design the mouse MITF construct. V.K. and A.Z. performed and analyzed RNA-seq experiments. C.L and J.H.H. supervised the project, interpretation of data and wrote the manuscript with input from all authors.

## Additional information

**Competing interests:** The authors declare no competing financial interests.

