## [Peer Review File · Nature Communications]

Reviewers' comments:

Reviewer #1 (Remarks to the Author):

In this manuscript, Sheinboim et al generate an inducible vector expressing MITF in ES cells and then use them to generate mice and differentiated cells and then go on to show that unlike in the differentiated cells, in the ES cells, MITF can not reprogram the cells to become melanocyte-like. However, the concept of Oct4 as a general inhibitor of differentiation-specific transcription factors is not new. Furthermore, there are multiple problems with the manuscript that need to be addressed. First, the authors overinterpret their results in numerous places and use overly fancy language to explain their results. This is best evidenced in their statement „...may serve as a gatekeeper protecting destabilization of Oct4 governed pluripotency by Mitf pro-differentiation activity.....in order to release their pluripotent stem cells from Oct4 pluripotency promoting and stabilizing function(s).“ Second, the statistical analysis of their bioinformatics data is not convincing. There is no statistics presented for the analysis presented in Figures 2d and 2e and the conclusion of the authors that the MEFs+Dox cells are similar to the primary melanocytes is not obvious. Similarly, they do not present statistical data showing that there is a likely increase in differentiation genes in the overlaps of Oct4 and the four transcription factors they characterized. In Figure 4a they show a significant overlap between Oct4 and Mitf peaks (38%) so 38% of differentiation genes are likely to pop up in their analysis. If they do the same for the other overlaps, are they seeing an overrepresentation of differentiation genes in all cases? Even so, one might ask whether that is important since these are differentiation genes, after all. They need to control for this by analyzing several other classes of genes to determine if they see a similar overlap. There are also questions about the definition of overlap. Is it too wide to allow 10kb +/- from the promoters or should they divide the peaks further, into enhancers and promoters and maybe restrict their analysis to +/- 2 or 5? Also, the Oct4 ChIP-seq analysis shows a lot of peaks so it is not surprising to see overlaps. Is the E2F7 transcription factor an appropriate control? Can other generic transcription factors also be used also as controls? Third, in most of their experiments, they induce Mitf expression for six days using Dox. This is a very long time and thus secondary effects are likely to be observed. Did they do a titration of Mitf expression on different days of cell culture? Why did they select 6 days for their analysis? Fourth, in order to verify the effects of MITF and OCT4 on the promoter constructs, they need to either mutate the binding sites in the reporter and show that the effects disappear, or use DNA-binding defective MITF/OCT4 constructs as controls. Fifth, Figure 5e is difficult to understand. This reviewer does not observe differences between the differentiation factors and other transcription factors, e.g. E2F7, from simply reading this chart. Sixth, previous analysis has shown that Oct4 interacts with a lot of different proteins. In order to verify the interactions with MITF, further studies would be recommended, including mapping of domains and verification of the interaction in additional cell lines and tissues.

Additional comments:

1. The abstract needs to be improved significantly.
2. In results, when describing the generation of the ES cells and mice, the terminology used to describe the constructs and cells needs to be improved and made consistent throughout (the name is not consistent with the name listed in the methods section). As it is currently, it is difficult to follow which lines are being used in the experiments. For example, in the first section describing the generation of the cells, they are called Col1a1-TetO-Mitf, whereas in the following section they are called KH2-MitfESCs. This is confusing.
3. The Southern blot in Figure 1b is not convincing.
4. In Figure 2b, why are only some cells MITF positive upon Dox treatment? Are these the same cells that are shown as Tyrp1 positive in the adjacent panel? If so, why are more cells Tyrp1 positive than are Mitf positive? Where is the staining for mCherry?
5. In the quantitation of the data shown in Figure 2b, it is not clear what the reference is. nCherry? Also, the „left panel“ and „right panel“ distinctions in these figure is not clear since the right panel is so small and appears only at the bottom.

6. The authors claim that their Mitf induction is: „...at an efficiency level of up to 100%“. This is an overstatement, at least judging from Figure 2b.
7. The authors claim that the high efficiency of transdifferentiation is due to „high levels of factor induction“. They can not claim this since they have not shown the level of Mitf protein in their system using western analysis.
8. In the section „Oct4 impedes Mitf pro-ESCs differentiation potential“ they authors say: „...further investigate our hypothesis in ESCs as well“. However they have not introduced a specific hypothesis at this point.
9. In this same section they „reasoned that ESCs will be favorable for manipulations, thereby enabling directed differentiation by Mitf induction“. However, this contradicts their earlier finding that Tyrosinase was not upregulated in the ESCs (Sup Fig 1b). The authors need to be more careful in their writing in order to avoid confusing the reader.
10. Later in the section they say „...it was unable to upregulate its specific target genes....(Fig. 3e). Figure 3e shows a reduction but not elimination of expression. The authors use too strong a language here.
11. Still later they say: „...Dox treated MEFs also inhibited Tyrosinase activity (Fig 3f)“. However, since they are not measuring activity they should say „expression“ instead of „activity“.
12. In the section „OCT4 interacts and interferes with MITF transcriptional activity“ they mention the ChIP-seq data for MITF. This was obtained from melanoma cells, not melanocytes as the authors state.
13. Do the UM3682 cells express Oct4 endogenously?
14. Later in the above section, the authors refer to Sup Fig 5b and say: „...were also found to be overexpressed apart from each other in skin cutaneous melanoma“. This is not what the figure shows. Rather, the figure shows a non-overlap in mutations in the two factors in melanomas.
15. Figure 4f needs improvement. If the HA antibody is used to bring down MITF, and both FLAG and HA antibodies used for staining the western, one would expect to get the same pattern as observed for the FLAG-OCT4 band in the input. However, the IP fraction shows a double band, similar to the HA-MITF band. Was there a mislabel of figures? What are the sizes of the expected bands?
16. The EMSA shown in Figure 4g is not acceptable. Nothing can reliably be read from this blot.
17. In Figure 4c, the contrast in the sequence needs to be improved to make it legible.
18. In the discussion, the authors start by stating that they have generated a superior system for MITF-mediated transdifferentiation. That is possible. However, this is an incremental improvement of our current knowledge and therefore of limited significance (also they say that they do this in a short period of 6 days – is that really short?). This part of they paper should be focused on establishing this as a system for analyzing differentiation factors.

--

Reviewer #2 (Remarks to the Author):

Sheinboim and Maza et al. report a method for achieving very much improved melanocyte trans-differentiation using a doxycycline-inducible transgenic mouse system. They also demonstrate that the pluripotency transcription factor OCT4 counteracts the melanocyte master regulator MITF and this inhibits melanocyte trans-differentiation. Furthermore, they postulate that OCT4 is actually a general factor clashing with different lineage master regulators through competitive binding to specific genomic loci, which creates a brake to trans-differentiation using pluripotent stem cells and potentially in other settings too. This work is very interesting and clearly explained/demonstrated, I believe it will stimulate further breakthrough discoveries in the field of trans-differentiation and it also has relevant implications for understanding pluripotency, development, and malignant transformation. I only have minor concerns:

1. In page 5, I think the enhanced conversion efficiency is not necessarily caused by higher expression levels of MITF in the transgenic cells but perhaps by optimized (could be neither too

high nor too low) and homogeneous levels. The authors may consider rephrasing this.

2. In page 5, when the authors talk of 100% trans-differentiation efficiency it may be convenient to add 'based on TYRP1 expression/immunofluorescence'.

3. In page 8, the heading says 'Oct4 prevents differentiation induced by multiple lineage commitment factors'. I would suggest using 'interferes' or similar rather than 'prevents', as the authors' data are very clear and suggestive but they don't provide additional experiments to demonstrate this. Also, can the authors postulate whether NANOG or SOX2 could be preventing other types of trans-differentiation/differentiation by binding to and competing with other types of lineage specifiers in the same way that OCT4 does with TITF?

4. Enhancers are important for cell identity maintenance. Besides promoter regions, is there any overlap at enhancer regions in the binding of OCT4 and MITF according to the ChIP-seq data?

5. Does OCT4 impede human somatic cell trans-differentiation as it does in mouse? The authors don't need to do this experiment, as this goes beyond the scope of the current study, but it would be nice to add one phrase containing speculation.

6. Be consistent with abbreviations/acronyms such as Hi-C throughout the text.

7. I also suggest revising gene versus protein terminology throughout the manuscript.

--

Reviewer #3 (Remarks to the Author):

Review of OCT4 impedes cell fate...

By Shienboim et al

This manuscript centers on the use of master regulators to differentiate and transdifferentiate mouse primary cells. The authors first characterize cells from different germ layers that have a specific transgene for inducible expression of Mitf.

Authors have found that while mouse embryonic fibroblasts can transdifferentiate into melanocytes at an efficiency of 100%, ESCs are refractory to such differentiation. They provide evidence that Oct4 expression is responsible for conferring resistance to differentiation.

Minor suggestions

Authors have focused on the interactions of tissue specific master regulators and Oct4 at the promoter regions, they should address whether such competitive binding also exists at the enhancer level.

Given the recent publication of the role of Oct4, Sox2 and Klf4 in enhancer occupancy and DNA binding at tissue specific DNA regulatory sequences, authors should reference the work by Chronis et al (Cell 168, 442-459) and thoroughly discuss their findings as they relate to Chronis's.

Reviewers' comments:

Reviewer #1 (Remarks to the Author):

In this manuscript, Sheinboim et al generate an inducible vector expressing MITF in ES cells and then use them to generate mice and differentiated cells and then go on to show that unlike in the differentiated cells, in the ES cells, MITF can not reprogram the cells to become melanocyte-like. However, the concept of Oct4 as a general inhibitor of differentiation-specific transcription factors is not new. Furthermore, there are multiple problems with the manuscript that need to be addressed. First, the authors overinterpret their results in numerous places and use overly fancy language to explain their results. This is best evidenced in their statement „...may serve as a gatekeeper protecting destabilization of Oct4governed pluripotency by Mitf pro-differentiation activity.....in order to release their pluripotent stem cells from Oct4 pluripotency promoting and stabilizing function(s).“

As suggested by the reviewer, we have carefully revised the manuscript editing or removing over-interpretations including the one highlighted by the reviewer.

Second, the statistical analysis of their bioinformatics data is not convincing. There is no statistics presented for the analysis presented in Figures 2d and 2e and the conclusion of the authors that the MEFs+Dox cells are similar to the primary melanocytes is not obvious.

We have followed the reviewer's suggestion and have added two new figures (**new Figures 2d and 2e**), and present the statistics in two new tables (**new Supplementary Table 1 and 2**); further we now provide detailed descriptions of the bioinformatics procedure in the methods and figure legends. In **new Figure 2d** we present a clustergram of 5864 genes differentially expressed (>4 fold difference) between MEFs and melanocytes. Expression is normalized to z score per row (per gene). We observed three main clusters: one corresponding to genes with low expression in MEF and high expression in primary melanocytes and DOX-treated MEFs, one corresponding to genes with high expression in MEFs and low in primary melanocytes and DOX-treated MEFs. This suggests that DOX induction causes a transition that changes the gene expression profile of MEFs to be more similar to that of melanocyte expression profile. In **new Figure 2e** we present Spearman correlation coefficients that demonstrate clearly that DOX-treated MEFs are more similar to primary melanocytes (0.26) than untreated MEFs (0.048), but overall DOX-treated MEFs are more similar to MEFs (0.73) than to primary melanocytes. Indicating that Dox+ is MEF that started to move towards primary melanocytes, but on the imaginary differentiation trajectory it is still closer to MEF. The statistical data shown in **new Supplementary Tables 1 and 2**.

Similarly, they do not present statistical data showing that there is a likely increase in differentiation genes in the overlaps of Oct4 and the four transcription factors they characterized.

Further, the statistics for Figure 5a have being added to the text: "-log10 (P-value): MITF (222.53); HES1 (177.38); GATA4 (235.73); CDX2 (193.11)".

In Figure 4a they show a significant overlap between Oct4 and Mitf peaks (38%) so 38% of differentiation genes are likely to pop up in their analysis. If they do the same for the other overlaps, are they seeing an overrepresentation of differentiation genes in all cases? Even so, one might ask whether that is important since these are differentiation genes, after all. They need to control for this by analyzing several other classes of genes to determine if they see a similar overlap.

We have followed the reviewer's suggestion and performed a new analysis using MITF ChIP-seq data generated in normal melanocytes in order to further analyze the overlap of MITF peaks with those of OCT4, E2F7, and P53, followed by GO enrichment analysis of the overlapped genes (**new Supplementary Table 6**). Consistent with our hypothesis, only the genes bound by MITF and OCT4 were known to function in melanocyte differentiation (pigmentation); in the overlaps with E2F7 or P53 no differentiation genes were observed. As shown in the original Figure 5a-b, only the lineage-specific genes were significantly enriched in the overlap with OCT4, and these genes only appear in relation to the lineage-specific transcription factor. For example, gut development related

genes were significant only in the OCT4 and CDX2 overlap, whereas the OCT4 overlap with MITF, HES1 or GATA4 did not contain gut-related genes.

There are also questions about the definition of overlap. Is it too wide to allow 10kb +/- from the promoters or should they divide the peaks further, into enhancers and promoters and maybe restrict their analysis to +/- 2 or 5? Also, the Oct4 ChIP-seq analysis shows a lot of peaks so it is not surprising to see overlaps.

We apologize for the unclear description of our analysis. Promoters in Figure 5a were defined as 1000 bp upstream to the TSS of a given gene. We evaluated overlap in this region. Figure 5b shows OCT4 peak overlap with MITF peaks along 10,000 bp. These figures clearly demonstrate that most of OCT4 peaks are at approximately 1500 bp from MITF peaks. We have followed the reviewer's suggestion and analyzed 1000 bp upstream and 500 bp downstream, 2000 bp upstream and 500 bp downstream, and 2000 bp upstream and 0 bp downstream of the TSS. The results are consistent as seen in the following table:

TSS upstream	TSS downstream	% of MITF peaks in promoters	% of MITF promoters with Oct4 peaks	% of MITF promoters close to Oct4 peaks	% of MITF promoters with no Oct4 peak
1000	0	11.5825026773	12.7458120903	26.0742898762	61.1798980335
1000	500	19.5897520389	15.2315015145	22.2414539161	62.5270445695
2000	500	21.056100173	16.9657422512	21.0032626427	62.030995106
2000	0	13.0488508114	15.5365371955	23.9631336406	60.5003291639

As suggested by the reviewer we have divided the peaks further into promoter-centered peaks (**new Figure 4a**) and enhancer-centered peaks. We have also analyzed the enhancers occupied by MITF and determined whether OCT4 and P53 co-binding these regions. We counted the number of MITF, OCT4, and P53 ChIP-seq peaks within embryonic stem cell (ESC) enhancers (as defined by ENCODE:

<http://hgdownload.cse.ucsc.edu/goldenPath/hg19/encodeDCC/wgEncodeBroadHm/>). Overall we found that 1.6% of enhancers are bound by MITF and 11% of MITF-bound enhancers are also bound by OCT4 (-log(p-value)=122). In comparison, 0.57% of MITF-bound enhancers are also bound by P53 (-log(p-value)=5). These data are now presented in **new Figure 5g** and in **new Supplementary Table 8**. These results suggest that the regulatory interaction between MITF and OCT4 on gene expression occurs in enhancers and in promoters. This strengthens our hypothesis that there is a transcriptional regulatory competition between MITF and OCT4.

Is the E2F7 transcription factor an appropriate control? Can other generic transcription factors also be used also as controls?

To address this comment, we performed a new analysis using MITF ChIP-seq data generated in normal melanocytes. We analyzed MITF peaks that overlapped with OCT4, E2F7, or P53, followed by GO enrichment analysis of the genes bound. **New Supplementary Table 6** clearly demonstrates that only the genes bound by both MITF and OCT4 showed significant melanocyte differentiation pathways (pigmentation), whereas in the overlap with E2F7 or P53 no differentiation pathways were observed. This strengthens our hypothesis that OCT4 specifically occupies genomic regions that are enriched with lineage-related pathways.

Third, in most of their experiments, they induce *Mitf* expression for six days using Dox. This is a very long time and thus secondary effects are likely to be observed. Did they do a titration of *Mitf* expression on different days of cell culture? Why did they select 6 days for their analysis?

Based on this question, we have treated MEF cells with DOX and analyzed MITF protein and mRNA levels at 6 h, 24 h, 48 h, 72 h, 96 h, and 144 h post induction. **New Supplementary Figure**

1b and c clearly demonstrate that MITF is greatly induced 6 hours post DOX treatment, however, MITF target genes peak at 6 days post induction.

Fourth, in order to verify the effects of MITF and OCT4 on the promoter constructs, they need to either mutate the binding sites in the reporter and show that the effects disappear, or use DNA-binding defective MITF/OCT4 constructs as controls.

Following the reviewer's suggestion we have used the mutated Oct4_T234E_S235E¹ as a control. We compared effects of wild-type OCT4 to that of a mutated OCT4 which was phosphorylated in a site located within the OCT4 homeobox domain (T234 and S235) and found to negatively regulate OCT4 by disrupting sequence-specific DNA binding¹. In **new Figure 4d** we demonstrate that WT Oct4 represses MITF transcriptional activation of *TRPM1* and *tyrosinase* significantly more efficiently than does mutated Oct4. This supports our hypothesis that OCT4 directly interferes with MITF transcriptional activity.

Fifth, Figure 5e is difficult to understand. This reviewer does not observe differences between the differentiation factors and other transcription factors, e.g. E2F7, from simply reading this chart.

We agree with the reviewer and have revised the text accordingly.

Sixth, previous analysis has shown that Oct4 interacts with a lot of different proteins. In order to verify the interactions with MITF, further studies would be recommended, including mapping of domains and verification of the interaction in additional cell lines and tissues.

We agree with the reviewer that further studies in additional cell lines and tissues will enhance our understanding of OCT4 interactions. This is beyond the scope of the present study and will be done in a follow-up work.

Additional comments:

1. The abstract needs to be improved significantly.

We have extensively modified the Abstract.

2. In results, when describing the generation of the ES cells and mice, the terminology used to describe the constructs and cells needs to be improved and made consistent throughout (the name is not consistent with the name listed in the methods section). As it is currently, it is difficult to follow which lines are being used in the experiments. For example, in the first section describing the generation of the cells, they are called Col1a1-TetO-Mitf, whereas in the following section they are called KH2-MitfESCs. This is confusing.

We have revised the Methods section and other sections of the paper in order to ensure consistent terminology throughout the paper.

3. The Southern blot in Figure 1b is not convincing.

We have now replaced the Southern blot with a version that has not been manipulated. Clone 3 shows only the 6.2-kb band, whereas clone 4 also shows an additional band at 4.1 kb. This indicates that clone 3 was not correctly targeted and only harbors the flope cassette; however, clone 4 was correctly targeted with teto-mitf as indicated by the 4.1-kb band.

4. In Figure 2b, why are only some cells MITF positive upon Dox treatment? Are these the same cells that are shown as Tyrp1 positive in the adjacent panel? If so, why are more cells Tyrp1 positive than are Mitf positive? Where is the staining for mCherry?

Most cells are MITF positive upon DOX treatment; however, the expression level varies. We have clarified this in the text and now describe in detail the results that indicate that the same cells expressing MITF co express Tyrp1. Importantly, Tyrp1 is a melanocyte-specific gene, involved in pigment production². Melanosomes are pigment-producing vesicles that are trafficked from melanocytes to adjacent cells³. Tyrp1 bound by a transmembrane domain to the melanosome membrane and is transported from melanocytes to neighboring cells². This transport explains the

Tyrp1 localization pattern in DOX-treated cells, as Tyrp1 appears in the cytoplasm and in neighboring cells where MITF levels are very low.

5. In the quantitation of the data shown in Figure 2b, it is not clear what the reference is. nCherry? Also, the „left panel“ and „right panel“ distinctions in these figure is not clear since the right panel is so small and appears only at the bottom.

We have generated a new figure demonstrating mCherry, MITF, and Tyrp1 co-localization in the same cell. **New Figure 2c** shows that in cells expressing nuclear mCherry, MITF appears in the nucleus and Tyrp1 in the cytoplasm. Additionally, the original figure was reorganized and moved to **Supplementary Figure 1e**. The methods and legends now describe how MITF and Tyrp1 expression intensities were measured and normalized to DAPI.

6. The authors claim that their Mitf induction is: „...at an efficiency level of up to 100%“. This is an overstatement, at least judging from Figure 2b.

We have modified the text according to the reviewer's suggestion.

7. The authors claim that the high efficiency of transdifferentiation is due to „high levels of factor induction“. They can not claim this since they have not shown the level of Mitf protein in their system using western analysis.

In the original version of the paper, Figure 2c demonstrates MITF protein levels 6 and 12 days post DOX treatment. We have now added **new Supplementary Figure 2b** that shows MITF protein levels at shorter times of DOX exposure.

8. In the section „Oct4 impedes Mitf pro-ESCs differentiation potential“ they authors say: „...further investigate our hypothesis in ESCs as well“. However they have not introduced a specific hypothesis at this point.

We have modified the text to explain our hypothesis prior to the section mentioned by the reviewer.

9. In this same section they „reasoned that ESCs will be favorable for manipulations, thereby enabling directed differentiation by Mitf induction“. However, this contradicts their earlier finding that Tyrosinase was not upregulated in the ESCs (Sup Fig 1b). The authors need to be more careful in their writing in order to avoid confusing the reader.

We have modified the text accordingly.

10. Later in the section they say „...it was unable to upregulate its specific target genes...(Fig. 3e). Figure 3e shows a reduction but not elimination of expression. The authors use too strong a language here.

We have revised the text to ensure that we do not overstate.

11. Still later they say: „...Dox treated MEFs also inhibited Tyrosinase activity (Fig 3f)“. However, since they are not measuring activity they should say „expression“ instead of „activity“.

We apologize for not clearly describing our experiment. In the original Figure 3f we measured tyrosinase activity as demonstrated by green Cy5 signal. We have modified the text to emphasize this point.

12. In the section „OCT4 interacts and interferes with MITF transcriptional activity“ they mention the ChIP-seq data for MITF. This was obtained from melanoma cells, not melanocytes as the authors state.

We have now analyzed ChIP-seq data from primary melanocytes to evaluate overlap of MITF peaks with E2F7, P53, and OCT4. **New Supplementary Table 6** includes the GO enrichment of the overlaps, clearly demonstrating that only in the OCT4-MITF overlap are GO terms for pigmentation genes significant.

13. Do the UM3682 cells express Oct4 endogenously?

The WM3682 melanoma cells express OCT4 at a low level, and OCT4 amounts are inversely correlated with MITF amounts, as seen in the original Supplementary Figure 5.

14. Later in the above section, the authors refer to Sup Fig 5b and say: "...were also found to be overexpressed apart from each other in skin cutaneous melanoma". This is not what the figure shows. Rather, the figure shows a non-overlap in mutations in the two factors in melanomas.

We agree with the reviewer and have modified the text accordingly.

15. Figure 4f needs improvement. If the HA antibody is used to bring down MITF, and both FLAG and HA antibodies used for staining the western, one would expect to get the same pattern as observed for the FLAG-OCT4 band in the input. However, the IP fraction shows a double band, similar to the HA-MITF band. Was there a mislabel of figures? What are the sizes of the expected bands?

As suggested, we have now added the complete IP analysis to the manuscript. MITF was immunoprecipitated (IP), and precipitate was analyzed for MITF and OCT4. This experiment, shown in **new Figure 4f**, demonstrates that OCT4 and MITF physically interact. Additionally, we have analyzed the OCT4 interaction with endogenous MITF by performing IP analysis in melanoma cells (**new Supplementary Figure 3c**).

16. The EMSA shown in Figure 4g is not acceptable. Nothing can reliably be read from this blot.

A new EMSA analysis was added. Revised **new Figure 4g** clearly demonstrates that OCT4 interfere with MITF DNA binding activity.

17. In Figure 4c, the contrast in the sequence needs to be improved to make it legible.

The contrast in Figure 4c has been improved.

18. In the discussion, the authors start by stating that they have generated a superior system for MITF-mediated transdifferentiation. That is possible. However, this is an incremental improvement of our current knowledge and therefore of limited significance (also they say that they do this in a short period of 6 days – is that really short?). This part of they paper should be focused on establishing this as a system for analyzing differentiation factors.

We agree with the reviewer and have modified the discussion accordingly.

--

Reviewer #2 (Remarks to the Author):

Sheinboim and Maza et al. report a method for achieving very much improved melanocyte trans-differentiation using a doxycycline-inducible transgenic mouse system. They also demonstrate that the pluripotency transcription factor OCT4 counteracts the melanocyte master regulator MITF and this inhibits melanocyte trans-differentiation. Furthermore, they postulate that OCT4 is actually a general factor clashing with different lineage master regulators through competitive binding to specific genomic loci, which creates a brake to trans-differentiation using pluripotent stem cells and potentially in other settings too. This work is very interesting and clearly explained/demonstrated, I believe it will stimulate further breakthrough discoveries in the field of trans-differentiation and it also has relevant implications for understanding pluripotency, development, and malignant transformation. I only have minor concerns:

1. In page 5, I think the enhanced conversion efficiency is not necessarily caused by higher expression levels of MITF in the transgenic cells but perhaps by optimized (could be neither too high nor too low) and homogeneous levels. The authors may consider rephrasing this.

We agree with the reviewer and have modified the discussion accordingly.

2. In page 5, when the authors talk of 100% trans-differentiation efficiency it may be convenient to add 'based on TYRP1 expression/immunofluorescence'.

The text has been revised: "The high efficiency of our DOX-inducible MITF system is likely due to an optimized transgenic delivery system of the reprogramming factor and to high levels of factor induction".

3. In page 8, the heading says 'Oct4 prevents differentiation induced by multiple lineage commitment factors'. I would suggest using 'interferes' or similar rather than 'prevents', as the authors' data are very clear and suggestive but they don't provide additional experiments to demonstrate this.

We agree with the reviewer and have modified the discussion as suggested.

Also, can the authors postulate whether NANOG or SOX2 could be preventing other types of trans-differentiation/differentiation by binding to and competing with other types of lineage specifiers in the same way that OCT4 does with TITF?

We apologize for not clearly describing our findings. In the original version of the manuscript, this analysis appeared in Supplementary Figure 8. As the reviewer suggested, NANOG and SOX are likely to compete with lineage specific transcription factors in similar way as OCT4. We have modified the text accordingly and have added two **new Supplementary Tables 9 and 10** that present the overlapping genes and GO analysis of NANOG and SOX2 with the lineage-specific transcription factors.

4. Enhancers are important for cell identity maintenance. Besides promoter regions, is there any overlap at enhancer regions in the binding of OCT4 and MITF according to the ChIP-seq data?

As suggested by the reviewer we have analyzed the enhancers occupied by MITF and determined whether OCT4 and P53 co-bind these regions. We counted the number of MITF, OCT4, and P53 ChIP-seq peaks within ESC enhancers (as defined by ENCODE: <http://hgdownload.cse.ucsc.edu/goldenPath/hg19/encodeDCC/wgEncodeBroadHmm/>). Overall we found that 1.6% of enhancers are bound by MITF and 11% of MITF-bound enhancers are also bound by OCT4 ($-\log(p\text{-value})=122$), whereas only 0.57% of MITF-bound enhancers are also bound by P53 ($-\log(p\text{-value})=5$). These data are now presented in **new Figure 5g** and in **new Supplementary Table 8**. These results suggest that regulatory interactions between MITF and OCT4 on gene expression occur in both enhancers and promoters. These data provide confirmation of our hypothesis that there is a transcriptional regulatory competition between MITF and OCT4.

5. Does OCT4 impede human somatic cell trans-differentiation as it does in mouse? The authors don't need to do this experiment, as this goes beyond the scope of the current study, but it would be nice to add one phrase containing speculation.

As suggested by the reviewer, we have added the following to the Discussion: "It would be interesting to validate a potential similar role for OCT4 in human somatic cell reprogramming".

6. Be consistent with abbreviations/acronyms such as Hi-C throughout the text.

We have revised the text to ensure that abbreviations and acronyms are used consistently.

7. I also suggest revising gene versus protein terminology throughout the manuscript.

We have carefully proofread the manuscript to make sure that we use proper gene vs. protein terminology.

--

Reviewer #3 (Remarks to the Author):

Review of OCT4 impedes cell fate...

By Shienboim et al

This manuscript centers on the use of master regulators to differentiate and transdifferentiate mouse primary cells. The authors first characterize cells from different germ layers that have a specific transgene for inducible expression of Mitf.

Authors have found that while mouse embryonic fibroblasts can transdifferentiate into melanocytes at an efficiency of 100%, ESCs are refractory to such differentiation. They provide evidence that Oct4 expression is responsible for conferring resistance to differentiation.

Minor suggestions

Authors have focused on the interactions of tissue specific master regulators and Oct4 at the promoter regions, they should address whether such competitive binding also exists at the enhancer level.

As suggested by the reviewer we have analyzed the enhancers occupied by MITF and determined whether OCT4 and P53 co-bind these regions. We counted the number of MITF, OCT4, and P53 ChIP-seq peaks within ESC enhancers (as defined by ENCODE: <http://hgdownload.cse.ucsc.edu/goldenPath/hg19/encodeDCC/wgEncodeBroadHmm/>). We found that 1.6% of enhancers are bound by MITF and 11% of MITF-bound enhancers are also bound by OCT4 (-log(p-value=122)). In contrast, only 0.57% of MITF-bound enhancers are also bound by P53 (-log(p-value=5)). These data are presented in new **Figure 5g** and in new **Supplementary Table 8**. This data suggest that the regulatory interaction between MITF and OCT4 on gene expression, is also true for enhancers and in promoter. These experiments confirm that there is a transcriptional regulatory competition between MITF and OCT4.

Given the recent publication of the role of Oct4, Sox2 and Klf4 in enhancer occupancy and DNA binding at tissue specific DNA regulatory sequences, authors should reference the work by Chronis et al (Cell 168, 442-459) and thoroughly discuss their findings as they relate to Chronis's.

As suggested by the reviewer, we now discuss the findings of Chronis et al. in the Discussion section: "The role of Oct4 in reprogramming of somatic cells into pluripotent stem cells has been widely studied (Takahashi and Yamanaka, 2006). Recently, Chronis et al. showed that an important step in this process is the silencing of MEF enhancers and promoters by Oct4, Sox2, Klf4, and cMyc. Interestingly, these TFs induce the relocation of somatic TFs from the MEF enhancers, emphasizing their role in blocking somatic TF-dependent differentiation."

References:

- 1 Brumbaugh, J. et al. Phosphorylation regulates human OCT4. Proceedings of the National Academy of Sciences of the United States of America 109, 7162-7168, doi:10.1073/pnas.1203874109 (2012).
- 2 Sarangarajan, R. & Boissy, R. E. Tyrp1 and oculocutaneous albinism type 3. Pigment cell research 14, 437-444 (2001).
- 3 Dror, S. et al. Melanoma miRNA trafficking controls tumour primary niche formation. Nature cell biology 18, 1006-1017, doi:10.1038/ncb3399 (2016).

Reviewers' comments:

Reviewer #1 (Remarks to the Author):

The authors have addressed most of my previous concerns and the manuscript is much improved in clarity and presentation. There are still some questions that need to be addressed. First, the co-IP studies indicate that MITF and OCT4 physically interact. However, according to the ChIP-seq peaks in Fig 4C, the sites are located quite a distance from each other. So are the authors suggesting long-range interactions? Also, in figure 4C they indicate MITF and OCT4 binding sites. Are these the same as the sites in the ChIP-seq drawing or are these different? According to the numbering of the binding sites this sequence is not contiguous. This needs to be made more clear, preferably by indicating genomic position. The EMSA blot is not convincing (Fig 4g). The model proposed for how OCT4 mediates its effects on MITF is not clear. The did not perform the final experiment of overexpressing MITF and at the same time siOCT4 in ES cells. According to the model, this should increase the efficiency of melanocyte differentiation.

Minor comments:

1. The sentence "This suggests that OCT4 protects against destabilization pluripotency potential, by..." is unclear. The authors presumably mean to say: "This suggests that OCT4 blocks MITF pro-differentiation activity."
2. In Figure 1, ATG needs to be explained. Also, the location of the DOX element should be indicated.

Reviewer #2 (Remarks to the Author):

The authors have done an excellent job in revising their manuscript, I don't have further comments.

Reviewer #3 (Remarks to the Author):

None.

Reviewers' comments:

Reviewer #1 (Remarks to the Author):

The authors have addressed most of my previous concerns and the manuscript is much improved in clarity and presentation. There are still some questions that need to be addressed.

We thank the reviewer for the through review and we are pleased we were able to address his/her concerns.

First, the co-IP studies indicate that MITF and OCT4 physically interact. However, according to the ChIP-seq peaks in Fig 4C, the sites are located quite a distance from each other. So are the authors suggesting long-range interactions?

Text was modified accordingly.

Also, in figure 4C they indicate MITF and OCT4 binding sites. Are these the same as the sites in the ChIP-seq drawing or are these different? According to the numbering of the binding sites this sequence is not contiguous. This needs to be made more clear, preferably by indicating genomic position.

The illustration of MITF and OCT4 ChIP-seq peaks was generated by uploading MITF and OCT4 ChIP-seq data^{1,2} into UCSC. We have now added a clarification in the Figure 4c legend. Furthermore, the sequences which appear in the original figure and include MITF and OCT4 binding elements, are the known promoters we have used at the luciferase reporter assay. To avoid misunderstanding, as suggested by the reviewer, we have now moved this panel to Figure 4d- which demonstrate the data of the luciferase promoter reporter.

The EMSA blot is not convincing (Fig 4g).

We have now added a graph which represent bands quantification from 3 independent EMSA experiments (**new Figure 4g**). A significant (** = $p < 0.05$) decrease in MITF binding to its promoter was observed, at the presence of OCT4.

The model proposed for how OCT4 mediates its effects on MITF is not clear. The did not perform the final experiment of overexpressing MITF and at the same time siOCT4 in ES cells. According to the model, this should increase the efficiency of melanocyte differentiation.

This is an important comment, however, we have done the suggested experiment. At the original version of the paper we have validated knocked down OCT4 using Lenti virus of sh-OCT4 or sh-Scrambled. MITF was then overexpressed in these treated ES cells by Dox addition (experimental design scheme in Figure 3g right panel). As suggested by the reviewer, more efficient differentiation of ES cells into melanocytes was demonstrated when OCT4 was knocked down (Figure 3g left panel).

Minor comments:

1. The sentence "This suggests that OCT4 protects against destabilizatin pluripotency potential, by..." is unclear. The authors presumably mean to say: "This suggests that OCT4 blocks MITF pro-differentiation activity."

Text was edit accordingly.

2. In Figure 1, ATG needs to be explained. Also, the location of the DOX element should be indicated.

In the target locus on the Col1a the Hygro resistance lacks an ATG initiation codon and therefore the cells are not Hygro resistant. However, the Mitf carrying flip-in construct has an ATG codon next to the Frt site. Therefore, cells that are correctly targeted acquire an ATG initiation codon and become Hygro resistant for cell culture selection.

The location of rtTA and Dox elements are now added to **new Figure 1a**.

Reviewer #2 (Remarks to the Author):

The authors have done an excellent job in revising their manuscript, I don't have further comments.

Reviewer #3 (Remarks to the Author):

None.

We are happy we satisfied both reviewers #2 and #3 and thank them for the constructive review which has certainly improved this manuscript.

References:

- 1 Strub, T. *et al.* Essential role of microphthalmia transcription factor for DNA replication, mitosis and genomic stability in melanoma. *Oncogene* **30**, 2319-2332 (2011).
- 2 Gifford, C. A. *et al.* Transcriptional and epigenetic dynamics during specification of human embryonic stem cells. *Cell* **153**, 1149-1163, doi:10.1016/j.cell.2013.04.037 (2013).

REVIEWERS' COMMENTS:

Reviewer #1 (Remarks to the Author):

The authors have addressed all my concerns.